# A Spitting Image:
## Superpixel Tokenization in Vision Transformers

## Abstract

Vision Transformer (ViT) architectures traditionally employ a rigid, grid-based approach to tokenization. In this work, we propose a generalized superpixel transformer (SPiT) framework that decouples tokenization from feature extraction; a significant shift from contemporary approaches, where these are treated as an undifferentiated whole. Using on-line superpixel tokenization and scale- and shape-invariant feature extraction, we perform experiments that contrast our approach with canonical tokenization and randomized partitions. We find that modular superpixel-based tokenization not only significantly enhances interpretability, as measured by state-of-the-art metrics for faithfulness, but also maintains competitive performance in classification. Our approach also demonstrates state-of-the-art capability in unsupervised salient segmentation, providing a space of semantically-rich models that can generalize across different vision tasks.

## 1 Introduction

ViTs (Dosovitskiy et al., 2021) have recently become the cynosure of vision tasks, outperforming convolutional architectures (CNNs). In the original transformer for language modeling (Vaswani et al., 2017; Devlin et al., 2019), *tokenization* serves as a crucial preprocessing step, with the aim of optimally dividing the data based on a predetermined entropic measure (Sennrich et al., 2016; Johnson et al., 2017). As transformers were adapted to vision, tokenization was simplified to partitioning of images into regular square grids. This approach proved remarkably effective (Liu et al., 2021; Touvron et al., 2021b;a; 2022; 2021c; Carion et al., 2020), and soon became canonical; an integral part of the architecture. Since their introduction, vision transformers have been shown to demonstrate inherent interpretability (Caron et al., 2021; Oquab et al., 2023), and their tokens can be leveraged for dense prediction tasks (Hamilton et al., 2022; Amir et al., 2022). However, square partitions incur a loss of resolution in the patch representation and subsequently do not inherently capture the resolution of the original images. For high-resolution dense predictions, images must be upscaled unless a separate decoder is used (Xie et al., 2021; Kirillov et al., 2023).

### 1.1 Motivation

We posit that the approach to tokenization in ViTs, while practical, glosses over the nuanced variability inherent in visual data. The uniformity imposed by square grid partitioning disregards the heterogeneity of semantic content across an image, resulting in token representations that are not aligned with image content and lack pixel-level granularity. Hence, exploring alternative tokenization methods is essential in order to further research into transformers for vision tasks.

We take a step back from the original ViT architecture to re-evaluate the role of the tokenizer in vision transformers and take inspiration from the success of language models (Brown et al., 2020; Ouyang et al., 2022) where tokenization is decoupled from the transformer backbone. We find a versatile analogy in *superpixels* which partition the image according to content, allowing for greater adaptability in scale and shape while effectively incorporating the semantic information inherent in visual content. The alignment of superpixels with semantic structures in images (Stutz et al., 2018) provides a compelling rationale for their integration into vision transformer architectures. This forms the foundation of our proposed method designed to address the inherent limitations of using square partitions as a minimal discrete unit for vision tasks and provides a framework for exploring alternative tokenization independent of architecture, illustrated in Fig. 1.

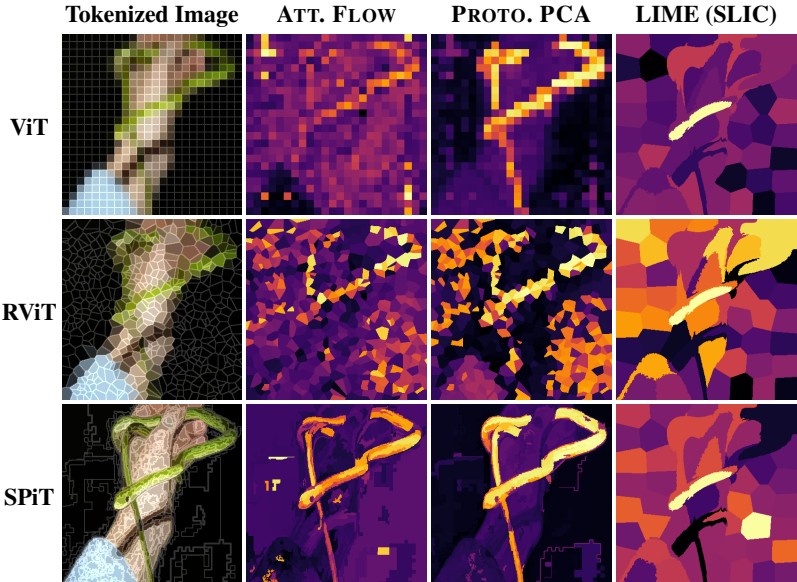

**Figure 1:** Visualization of tokens and feature attributions for prediction "grass snake" with different tokenization strategies: square tokens (ViT), random Voronoi tokens (RViT) and superpixel tokens (SPiT). See more visualizations in Fig. D.1.

## 1.2 CONTRIBUTIONS

Our research prompts two specific inquiries: **(a)** *Is a rigid adherence to square patches necessary?*, and **(b)** *What advantages might alternative tokenization strategies unveil?* We establish that:

- **Generalized Framework:** Superpixel tokenization is not orthogonal to canonical ViT architectures, but instead generalizes them in a modular scheme, providing a richer space of transformers for vision tasks where *the transformer backbone is independent of our tokenization framework*.
- **Efficient Tokenization:** We introduce an efficient on-line tokenization approach with pixel-level granularity, complemented by feature extraction independent of rigid patch sizes. It incorporates learnable positional encodings informed by shape and scale, adaptable to irregular patches. Our framework achieves training and inference times comparable to standard ViT models, while maintaining classification performance.
- **Refined Spatial Resolution:** Superpixel tokens provide more granularity in spatial resolution more semantically aligned with image content. We demonstrate that our framework achieves SotA results in unsupervised salient segmentation, and conduct experiments that show that our framework yields more faithful attributions compared to established explainability methods.

Our main contribution is *the introduction of a novel way of thinking about tokenization*, which has been overlooked in recent works—cf. our discussion in Section 4. We propose a modular and scalable framework for decoupling tokenization from the transformer backbone in standard ViT architectures. Our experiments establish a fair comparison against well-known baselines without the inclusion of various optimizations to the backbone (i.e., vanilla ViT architectures), highlighting the importance of understanding tokenizers for different tasks. This controlled comparison is crucial for attributing observed performance disparities specifically to the tokenization techniques under scrutiny, and eliminates confounding factors from specialized architectures or training regimes. Hence, our focus is on *understanding the impact of irregular tokenization in ViTs* instead of optimizing models for a particular downstream task.

**Preliminary Notation:** We let $H \times W = \{(y, x) : 1 \leq y \leq h, 1 \leq x \leq w\}$ denote the coordinates of an image of spatial dimension $(h, w)$, and let $\mathcal{I}$ be an index set for the mapping $i \mapsto (y, x)$. This allows us to consider a $C$-channel image as a signal $\xi \colon \mathcal{I} \to \mathbb{R}^C$. In general, we assume $C = 3$. At times, we will leverage a tensor representation denoted by $\vec{\xi} \in \mathbb{R}^{H \times W \times C}$, and we use the capitalized $\Xi$ to denote extracted features of various modalities. We use the general vectorization operator $\mathrm{vec} \colon \mathbb{R}^{d_1 \times \cdots \times d_n} \to \mathbb{R}^{d_1 \ldots d_n}$, and denote function composition by $f(g(x)) = (f \circ g)(x)$.

## 2 METHODOLOGY: MODULAR TOKENIZATION IN VISION TRANSFORMERS

We set out to design a framework for *modular tokenization*, independent of the transformer architecture, with the following incentives. Firstly, the ability to transfer knowledge and adapt models to new tasks or data is a crucial component in advancing research in deep learning. Modular tokenization enables precisely this by allowing models to interchange tokenization strategies based on the requirements of different datasets or tasks. This flexibility not only enhances the utility of pre-trained models and accelerates the development cycle of deploying models to new applications; but also provides the opportunity to swap or optimize tokenization strategies without the need for overhauling the entire model architecture.

Secondly, we note that different visual problems could require distinct approaches to tokenization. This choice influences how the model perceives and interacts with visual data, shaping its ability to capture features, relationships, and contextual information. A modular framework allows for the application of tailored tokenization strategies that align optimally with the specific demands of each problem, and potentially allows us to embed *visual priors* into the modeling process. Modular tokenization constitutes a powerful tool for analyzing models and studying components in isolation. Combined with the granularity of superpixels, this allows for extracting interpretations and explanations at the pixel-level, in response to increased demands for transparency in vision models.

### 2.1 FRAMEWORK

We generalize the canonical ViT architecture by allowing for a modular tokenizer and different methods of feature extraction. Note that a canonical ViT is generally presented as a three-component system with a tokenizer-embedder $g$, a backbone $f$ consisting of a sequence of attention blocks, and a subsequent prediction head $h$. Contrarily, NLP transformers explicitly decouples $g$ from the backbone $f$. Following this lead, we note that we can essentially rewrite a embedding module as a three component modular system, featuring a tokenizer $\tau$, a feature extractor $\phi$, and an embedder $\gamma$ such that $g = \gamma \circ \phi \circ \tau$, emphasizing that these are inherent components in the original architecture, but "hidden" by a simplified tokenization

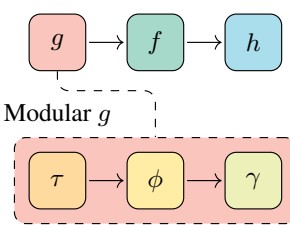

**Figure 2:** Illustration of modular tokenization in ViT architecture.

strategy. This framework allows for a more complete assessment of the model as a five component feedforward system on the form

$$\Phi(\vec{\xi}; \theta) = (h \circ f \circ g)(\vec{\xi}; \theta), \tag{1a}$$

$$= (h \circ f \circ \gamma \circ \phi \circ \tau)(\vec{\xi}; \theta), \tag{1b}$$

where $\theta$ denotes the set of learnable parameters of the model, and $\circ$ denotes function composition. Figure 2 illustrates the modular view on $g$. In a standard ViT model, the tokenizer $\tau$ acts by partitioning the image into fixed-size square partitions. This directly provides vectorized features since patches are of uniform dimensionality and ordering, hence $\phi = \text{vec}$ in standard ViT architectures. The embedding $\gamma$ is typically a learnable linear layer, mapping features to the embedding dimension of the specific architecture. The feature extractor $\phi$ distinguishes transformer architectures for images from those for language. In language models, symbolic characters and words are numerically represented solely through embeddings. Contrarily, image data is inherently numeric, necessitating an extraction mechanism to generate features, particularly in the case of irregular partitions.

### 2.2 PARTITIONING AND TOKENIZATION

Tokenization in language tasks fundamentally involves partitioning characters or words into discrete segments. Mirroring this procedure in vision tasks induces a comparable discrete partitioning of a digital image. For a data driven partitioning task with raw pixel features, we construct a hierarchical parallel on-line superpixel tokenizer. Our approach is similar to the method outlined by Wei et al. (2018), but differs by an explicit regularization term and the fact that the aggregation is computed in parallel with customized CUDA kernels over the full image graph at each step $t$. We also leverage anisotropic diffusion as a preprocessing step over the features used for computing the superpixel hierarchy, shown to be effective in the work by Xiaohan et al. (1992). We contrast our proposed

superpixel tokenized model (SPiT) to two alternate approaches, the canonical square tokenization in standard ViTs, and random Voronoi tessellations (RViT) selected for being well defined mathematical objects commonly used to generate random tilings of the plane. See Fig. 1's first column for examples of the different tokenization schemes.

**Superpixel Graphs**: Let $E^{(0)} \subset \mathcal{I} \times \mathcal{I}$ denote the four-way adjacency edges under $H \times W$. We consider a superpixel as a set $S \subset \mathcal{I}$, and we say that $S$ is connected if for any two pixels $p, q \in S$, there exists a sequence of edges in $\left( (i_j, i_{j+1}) \in E^{(0)} \right)_{j=1}^{k-1}$ such that $i_1 = p$ and $i_k = q$. A set of superpixels form a partition $\pi$ of an image if for any two distinct superpixels $S, S' \in \pi$, their intersection $S \cap S' = \emptyset$, and the union of all superpixels is equal to the set of all pixel positions in the image, i.e., $\bigcup_{S \in \pi^{(t)}} S = \mathcal{I}$.

Let $\Pi(\mathcal{I}) \subset 2^{2^{\mathcal{I}}}$ denote the space of all partitions of an image, and consider a sequence of partitions $(\pi^{(t)})_{t=0}^{T}$. We say that a partition $\pi^{(t)}$ is a refinement of another partition $\pi^{(t+1)}$ if for all superpixels $S \in \pi^{(t)}$ there exists a superpixel $S' \in \pi^{(t+1)}$ such that $S \subseteq S'$, and we write $\pi^{(t)} \sqsubseteq \pi^{(t+1)}$. Our goal is to construct a $T$-level hierarchical partitioning of the pixel indices $\mathcal{H} = \left( \pi^{(t)} \in \Pi(\mathcal{I}) : \pi^{(t)} \sqsubseteq \pi^{(t+1)} \right)_{t=0}^{T}$ such that each superpixel is connected.

To construct $\mathcal{H}$, the idea is to successively join vertices by parallel edge contraction to update the partition $\pi^{(t)} \mapsto \pi^{(t+1)}$. We do this by considering each level of the hierarchy as a graph $G^{(t)}$ where each vertex $v \in V^{(t)}$ is the index of a superpixel in the partition $\pi^{(t)}$, and each edge $(u, v) \in E^{(t)}$ represent adjacent superpixels for levels $t = 0, \ldots, T$. The initial image can thus be represented as a grid graph $G^{(0)} = (V^{(0)}, E^{(0)})$ corresponding to the singleton partition $\pi^{(0)} = \left\{ \{i\} : i \in \mathcal{I} \right\}$.

**Weight function**: To apply the edge contraction, we first define an edge weight functional $w_{\xi}^{(t)} \colon E^{(t)} \to \mathbb{R}$. In the the method by Wei et al. (2018), self-loops are removed after edge contraction. We retain these loops, which we use to constrain the growth of larger superpixels by weighting loops by the size of the superpixel. This acts as a regularizer by constraining the variance of superpixel sizes, in contrast to the method proposed by Wei et al. (2018).

To this end, we define separate weight functionals for self-looping edges. We compute the empirical mean and standard deviation of the size of the superpixels for each level $t$ in the hierarchy, and denote these by $\mu_{|\pi|}^{(t)}, \sigma_{|\pi|}^{(t)}$, respectively. For the non-loops we use the averaged features of each superpixel $\mu_{\xi}^{(t)}(v) = \sum_{i \in \pi_v^{(t)}} \xi(i) / |\pi_v^{(t)}|$, and use cosine similarity given by

$$\text{sim}\left( \mu_{\xi}^{(t)}(u), \mu_{\xi}^{(t)}(v) \right) = \frac{\langle \mu_{\xi}^{(t)}(u), \mu_{\xi}^{(t)}(v) \rangle}{\|\mu_{\xi}^{(t)}(u)\| \cdot \|\mu_{\xi}^{(t)}(v)\|}. \tag{2}$$

The final edge weight function is then given by

$$w_{\xi}(u, v) = \begin{cases} \text{sim}\left( \mu_{\xi}^{(t)}(u), \mu_{\xi}^{(t)}(v) \right), & \text{for } u \neq v; \\ \left( |\pi_u^{(t)}| - \mu_{|\pi|}^{(t)} \right) / \sigma_{|\pi|}^{(t)}, & \text{otherwise.} \end{cases} \tag{3}$$

**Update rule**: We use a greedy parallel update rule for the edge contraction, such that each superpixel joins with a neighboring superpixel with the highest edge weights. We also note that we include self-loops for all $G^{(t)}$ where $t \geq 1$. Let $\mathfrak{N}^{(t)}(v)$ denote the neighborhood of adjacent vertices of the superpixel with index $v$ at level $t$. We construct an intermediate set of edges, given by

$$E^{(t+\frac{1}{2})} = \left( v, \arg\max_{u \in \mathfrak{N}^{(t)}(v)} w_{\xi}(u, v) : v \in V^{(t)} \right). \tag{4}$$

Now, the transitive closure $\overline{E^{(t+\frac{1}{2})}}$, i.e., the connected components of $E^{(t+\frac{1}{2})}$, explicitly yields a mapping $V^{(t)} \mapsto V^{(t+1)}$ such that $\pi_v^{(t+1)} = \bigcup_{u \in \overline{\mathfrak{N}^{(t+\frac{1}{2})}}(v)} \pi_u^{(t)}$, where $\overline{\mathfrak{N}^{(t+\frac{1}{2})}}(v)$ denotes the connected component of vertex $v$ in $\overline{E^{(t+\frac{1}{2})}}$. This update rule for the partitions ensures that each partition at level $(t + 1)$ is a connected region, as it is formed by merging adjacent superpixels with the highest edge weights. We show an illustration of this aggregation in Fig. 3.

**Iterative refinement**: We repeat the steps of computing aggregation maps, regularized edge weights, and edge contraction until the desired number of hierarchical levels $T$ is reached. At each level, the partitions become more coarse, representing larger homogeneous regions in the image. The hierarchical structure allows for a flexible multiscale representation of the image, capturing both local and global structures. At level $T$ we have obtained a sequence of partitions $(\pi^{(t)})_{t=0}^{T}$, where each partition at level $t$ is a connected region of the image, and $\pi^{(t)} \sqsubseteq \pi^{(t+1)}$ for all $t$.

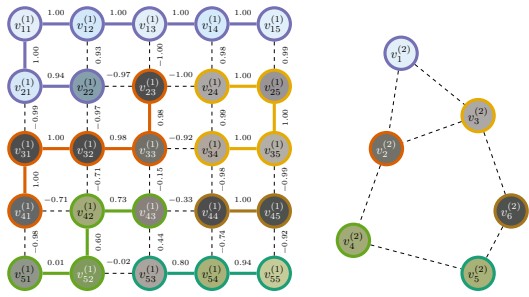

**Figure 3:** Visualization of superpixel aggregation.

We conduct experiments to empirically verify the relationship between the number of tokens produced by varying the steps $T$ and patch size $\rho$ in canonical ViT tokenizers. Let $N_{\text{SPiT}}, N_{\text{ViT}}$ denote the number of tokens for the SPiT tokenizer and ViT tokenizer respectively. Remarkably, we are able to show with a high degree of confidence that the relationship is $\mathbb{E}(T \mid N_{\text{SPiT}} = N_{\text{ViT}}) = \log_2 \rho$, *regardless of image size*. Results and more details can be found in Appendix F.

## 2.3 Feature Extraction with Irregular Patches

While we conjecture the choice of square patches in the ViT architecture to be motivated by convenience, it is naturally also a result of the challenge posed by the alternative. Irregular patches are unaligned, exhibit different shapes and dimensionality, and are not necessarily convex. These factors make the embedding of irregular patches to a common inner product space nontrivial. Which properties, then, do the features need to encode to represent irregular image patches? In addition to consistency and uniform dimensionality, we propose a minimal set of properties any such features would need to capture; *color, texture, shape, scale*, and *position*.

**Positional Encoding**: To encode shape, scale and position, we propose applying a joint histogram over the coordinates of a superpixel $S_n$ for each of the $n = 1, \ldots, N$ partitions. First, we normalize the positions such that $(y', x') \in [-1, 1]^2$ for all $(y', x') \in S_n$. We decide on a fixed number of bins $\beta$, denoting the dimensionality of our features in each spatial direction, and apply kernel density estimation to construct a joint histogram $\hat{\Xi}^{(\text{pos})} \in \mathbb{R}^{\beta \times \beta}$ using a Gaussian kernel $K_\sigma$ with bandwidth $\sigma \in [0.01, 0.05]$ such that

$$\hat{\Xi}_{n,y,x}^{(\text{pos})} = \sum_{(y_j, x_j) \in S_n} K_\sigma(y - y_j, x - x_j). \tag{5}$$

This, in effect, encodes the position of the patch within the image, as well as its shape and scale. Finally, spatial dimensions for each superpixel are flattened such that $\Xi_n^{(\text{pos})} = \text{vec}(\hat{\Xi}_n^{(\text{pos})}) \in \mathbb{R}^{\beta^2}$.

**Texture Features**: Gradient operators provides a simple but relatively robust method of extracting texture information (Leung & Malik, 2001). We use the gradient operator proposed by Scharr (2007) for this purpose due to it providing better rotational symmetry while minimizing discretization error, and we normalize the gradient operator over averaged channels such that $\Delta\xi \in [-1, 1]^{H \times W \times 2}$, where the last dimensions correspond to each gradient direction in the image. Mirroring the procedure for the positional features, we then construct a joint histogram with a Gaussian KDE kernel over the gradients within each superpixel $S_n$, giving the vectorized features $\Xi_n^{(\text{grad})} \in \mathbb{R}^{\beta^2}$.

While our proposed gradient features are commensurable with the canonical ViT architecture, they represent an additional dimension of information. We therefore perform ablations on the effect of including or omitting these features across architectures. For models where these features are omitted, we say that the feature extractor $\phi$ is *gradient excluding*.

**Color Features**: To encode the light intensity information from the raw pixel data into our features, we interpolate the bounding boxes of each patch to a fixed resolution of $\beta \times \beta$ using a bilinear interpolation operator, while masking out the pixel information in other surrounding patches. These features essentially capture the raw pixel information of the original patches, but resampled and scaled to uniform dimensionality. We call our proposed feature extractor $\phi$ an *interpolating feature*

**Table 1:** Model performance and accuracy (Top 1) on classification tasks.

| Model | | | | Perf. | | IN1K (384) | | INREAL (384) | | CIFAR100 (256) | | CALTECH256 (256) | |
|---|---|---|---|---|---|---|---|---|---|---|---|---|---|
| Name | Tok. | Feat. | Grad. | # Par. | Im./s.[‡] | KNN | Lin. | KNN | Lin. | KNN | Lin. | KNN | Lin. |
| ViT-S16 | Sqr. | Pix. | ✗ | 22.1M | — | 0.692 | 0.765 | 0.970 | 0.778 | 0.833 | 0.827 | 0.827 | 0.818 |
| ViT-S16 | Sqr. | Pix. | ✓ | 22.2M | — | 0.682 | 0.754 | 0.974 | 0.782 | 0.836 | 0.830 | 0.832 | 0.824 |
| RViT-S16 | Vor.[†] | Intp. | ✗ | 22.1M | — | 0.740 | **0.767** | **0.977** | **0.829** | 0.858 | 0.856 | 0.858 | 0.852 |
| RViT-S16 | Vor.[†] | Intp. | ✓ | 22.2M | — | **0.741** | 0.759 | 0.974 | 0.818 | **0.859** | **0.856** | **0.861** | **0.856** |
| SPiT-S16 | SPix. | Intp. | ✗ | 22.1M | — | 0.628 | 0.689 | 0.956 | 0.746 | 0.769 | 0.761 | 0.771 | 0.767 |
| SPiT-S16 | SPix. | Intp. | ✓ | 22.2M | — | 0.736 | 0.750 | 0.973 | 0.819 | 0.839 | 0.832 | 0.851 | 0.849 |
| ViT-B16 | Sqr. | Pix. | ✗ | 86.6M | 793.04 | 0.737 | 0.802 | 0.978 | 0.853 | 0.897 | 0.892 | 0.879 | 0.879 |
| ViT-B16 | Sqr. | Pix. | ✓ | 86.8M | 721.12 | 0.748 | **0.805** | 0.975 | 0.854 | **0.899** | **0.899** | 0.885 | **0.889** |
| RViT-B16 | Vor.[†] | Intp. | ✗ | 86.6M | 619.86 | 0.718 | 0.788 | 0.958 | 0.843 | 0.838 | 0.894 | 0.882 | 0.873 |
| RViT-B16 | Vor.[†] | Intp. | ✓ | 86.8M | 585.64 | 0.725 | 0.789 | 0.962 | 0.841 | 0.762 | 0.888 | 0.861 | 0.864 |
| SPiT-B16 | SPix. | Intp. | ✗ | 86.6M | 690.72 | 0.569 | 0.760 | 0.954 | 0.793 | 0.634 | 0.813 | 0.829 | 0.833 |
| SPiT-B16 | SPix. | Intp. | ✓ | 86.8M | 640.59 | **0.752** | 0.804 | **0.980** | **0.858** | 0.845 | 0.884 | **0.891** | 0.888 |

[†]Uncertainty measures for scores from the stochastic Voronoi (RViT) tokenizer are detailed in Appendix Table E.1.
[‡]Median throughput estimated over full training with $4\times$ MI250X GPUs using `float32` precision.

*extractor.* Similar to positional and texture features, the RGB features are normalized to $[-1, 1]$ and vectorized such that $\Xi^{(\mathrm{col})} \in \mathbb{R}^{3\beta^2}$. The feature modalities are concatenated to yield the final features $\Xi_n = [\Xi_n^{(\mathrm{col})}, \Xi_n^{(\mathrm{pos})}, \Xi_n^{(\mathrm{grad})}] \in \mathbb{R}^{5\beta^2}$. For gradient excluding feature extractors, the gradient features are dropped such that $\Xi_n \setminus \Xi_n^{(\mathrm{grad})} = [\Xi_n^{(\mathrm{col})}, \Xi_n^{(\mathrm{pos})}] \in \mathbb{R}^{4\beta^2}$.

### 2.4 GENERALIZATION OF CANONICAL VIT

Our proposed feature extraction framework essentially acts as a generalization of the canonical ViT framework, and is equivalent to applying an canonical tokenizer using a fixed patch size $\rho$ with interpolated feature extraction.

**Proposition 2.1** (Embedding Equivalence). *Let $\tau^*$ denote an canonical ViT tokenizer with a fixed patch size $\rho$, let $\phi$ denote a gradient excluding interpolated feature extractor, and let $\gamma^*, \gamma$ denote embedding layers with equivalent linear projections $L_\theta^* = L_\theta$. Let $\Xi^{(\mathrm{pos})} \in \mathbb{R}^{N \times \beta^2}$ denote a matrix of vectorized joint histogram positional embeddings under the partitioning induced by $\tau^*$. Then for $H = W = \beta^2 = \rho^2$, the embeddings given by $\gamma \circ \phi \circ \tau^*$ are equivalent to the canonical ViT embeddings given by $\gamma^* \circ \phi^* \circ \tau^*$ up to proportionality.*

We provide necessary definitions and proofs for Prop. 2.1 in the appendix, demonstrating that our proposed framework includes the canonical ViT architecture as a special case. This provides opportunities for transfer learning, where pre-trained models can be fine-tuned to leverage our proposed modular framework. The modularity of our framework can also be exploited to optimize components individually, facilitating continued analysis and understanding of transformers for vision tasks.

## 3 EXPERIMENTS

To evaluate our SPiT framework, we train models with small (S) and base (B) capacities on a general purpose classification task using IMAGENET1K (Deng et al., 2009), where the ViT serves as a baseline. We evaluate the models by fine-tuning on CIFAR100 (Krizhevsky et al., 2009) and CALTECH256 (Griffin et al., 2022), in addition to validation using the IMAGENET REAL labels (Beyer et al., 2020). We ablate over the inclusion of gradient features in different models. We also evaluated our models with a KNN classifier, which provides some insight into the properties of the embedding space for the different models. Table 1 gives an overview of the results. We present the details of our setup in Appendix C.

**Classification:** Our results demonstrate that ViTs can be successfully trained under irregular superpixel tokenization for classification tasks. For models with gradient texture features, superpixel tokenization performs comparably to square partitioning. We observe that models with irregular tokens and gradient excluding feature extractors underperform. We conjecture that this is likely due to irregularity and nonconvexity of superpixels, and largely confirms our conjecture that gradient features can compensate for loss of information from interpolation. Our findings in Section 2.4 also supports this.

**Table 2:** Results for unsupervised salient segmentation.

| TokenCut Backbone | Postproc. | ECSSD | | | DUTS | | | DUT-OMRON | | |
|---|---|---|---|---|---|---|---|---|---|---|
| | | max $F_\beta$ | IoU | Acc. | max $F_\beta$ | IoU | Acc. | max $F_\beta$ | IoU | Acc. |
| DINO (Wang et al., 2022b) | ✗ | 0.803 | 0.712 | 0.918 | 0.672 | 0.576 | 0.903 | 0.600 | 0.533 | 0.880 |
| | ✓ | 0.874 | 0.772 | **0.934** | 0.755 | 0.624 | **0.914** | 0.697 | **0.618** | **0.897** |
| SPiT | ✗ | **0.903** | **0.773** | **0.934** | **0.771** | **0.639** | 0.894 | **0.711** | 0.564 | 0.868 |

When comparing results across the different datasets, we unsurprisingly note stronger results on CIFAR100 for the ViT models. *This is to be expected*, since the original images are exceedingly low resolution (32) and have been upscaled (256). While we use bilinear interpolation for upscaling, the square artifacts from the low resolution still persist to some degree, and hence naturally align more with square tokenization. On the other hand, we were more surprised to see that SPiT performs better than the ViT over IMAGENET REAL, indicating that the model generalizes somewhat better. While certain results indicates that SPiT outperforms the baseline ViT, we stress that *the results are not significant enough to warrant any clear benefit for any framework in particular on general purpose classification tasks.* We note that *comparable performance is a positive result*, since our focus is on demonstrating the feasibility of modular superpixel tokenization as a new research direction for vision transformers. See more details in Appendix E.

**Unsupervised Salient Segmentation:** Superpixels have historically been applied in dense prediction tasks such as segmentation and object detection (Ladický et al., 2009; Yan et al., 2015) as a lower-dimensional prior for dense prediction tasks. To evaluate our framework, we are particularly interested in tasks for which the pretrained model can be leveraged directly to demonstrate the inherent benefits of the tokenizer. Wang et al. (2022b) propose an unsupervised methodology for extracting salient segmentation maps for any transformer model using normalized graph cut (Shi & Malik, 2000). We conduct experiments extending this well-established method to showcase preliminary out-of-the-box capabilities on dense prediction tasks for our proposed tokenizer, with details in Appendix G.

Table 2 shows results for the ECSSD (Yan et al., 2013), DUTS (Wang et al., 2017) and DUT-OMRON (Yang et al., 2013) datasets, and demonstrates that SPiT compares favorably to the application of DINO (Caron et al., 2021) under the TokenCut framework, notably without any form of postprocessing. The results clearly indicates that our tokenizer has stronger semantic alignment with image content, and that our proposed framework is capable of dense predictions without learnable tokenization. We use the same metrics as the original TokenCut framework; for max $F_\beta$ we set $\beta = 1/3$ and take the maximum score over 255 uniformly sampled thresholds. Visualization of a set of random results are featured in Fig. G.1.

## 3.1 EFFECT ON INTERPRETABILITY FROM REFINED ATTENTION MAPS

One of the salient features of transformers is the inherent interpretability provided by their attention mechanisms. Techniques such as attention rollout (Dosovitskiy et al., 2021), attention flows (Abnar & Zuidema, 2020), class token attention maps (Caron et al., 2021), and PCA projections over common sets of images (Oquab et al., 2023) have been leveraged to visualize the reasoning behind the model's decisions. These techniques are constrained by the granularity and semantic alignment of the original tokenization scheme. In contrast, LIME (Ribeiro et al., 2016) provides a well-established framework for post-hoc explainability with superpixel partitions using Quickshift (Vedaldi & Soatto, 2008) or SLIC (Achanta et al., 2012) for local linear surrogate models.

Since superpixel tokenization naturally augment attention maps with a higher level of granularity, we conducted experiments to quantify the faithfulness of interpretations under different tokenization strategies. To evaluate the inherent interpretability we compute the attention flow of the model in addition to PCA projected features, infused with prototypes for the predicted class, and contrast this with attributions from LIME with independently computed SLIC superpixels. We measure faithfulness using *comprehensiveness* (COMP) and *sufficiency* (SUFF) (DeYoung et al., 2020), which have been shown to be the two strongest quantitative measures for transformers (Chan et al., 2022). See Appendix D for experimental details.

**Table 3:** Faithfulness of Attributions, w. CI (95%).

| | ViT-B16 (IN1K) | | RViT-B16 (IN1K) | | SPiT-B16 (IN1K) | |
| | COMP ↑ | SUFF ↓ | COMP ↑ | SUFF ↓ | COMP ↑ | SUFF ↓ |
|---|---|---|---|---|---|---|
| LIME/SLIC | **0.244 ± 0.004** | **0.543 ± 0.006** | **0.236 ± 0.004** | **0.591 ± 0.007** | 0.244 ± 0.005 | **0.520 ± 0.006** |
| ATT.FLOW | 0.160 ± 0.004 | 0.664 ± 0.006 | 0.223 ± 0.005 | 0.685 ± 0.007 | **0.259 ± 0.006** | 0.558 ± 0.006 |
| PROT.PCA | 0.206 ± 0.005 | 0.710 ± 0.006 | 0.209 ± 0.005 | 0.691 ± 0.007 | 0.256 ± 0.005 | 0.592 ± 0.006 |

**Color coding:** baseline, weaker than baseline, stronger than baseline.

**Table 4:** Tokenizer Generalization.

| | | Δ Acc. ↑ (IN1K) | | |
| Model | Grad. | Sqr. | Vor. | SPix. |
|---|---|---|---|---|
| ViT-B16 | ✗ | 0.000 | −0.551 | −0.801 |
| ViT-B16 | ✓ | 0.000 | −0.494 | −0.798 |
| RViT-B16 | ✗ | **0.006** | 0.000 | −0.593 |
| RViT-B16 | ✓ | 0.003 | 0.000 | **−0.163** |
| SPiT-B16 | ✗ | −0.407 | −0.464 | 0.000 |
| SPiT-B16 | ✓ | −0.200 | **−0.063** | 0.000 |

**Table 5:** Superpixel Quality and Efficiency.

| | BSDS500 | | SBD[‡] | | Runtime |
| | $\mathbb{V}_{\text{Expl.}}$↑ | $|\pi|$ ↓ | $\mathbb{V}_{\text{Expl.}}$ ↑ | $|\pi|$ ↓ | sec/img. ↓ |
|---|---|---|---|---|---|
| ETPS[†] | 0.924 | 651.0 | 0.955 | 648.1 | 0.3268 |
| SEEDS[†] | 0.901 | 670.6 | 0.944 | 644.9 | 0.4501 |
| SLIC[†] | 0.847 | 575.3 | 0.897 | 592.2 | 0.0729 |
| Watershed[†] | 0.803 | 608.1 | 0.871 | 641.1 | 0.0038 |
| SPiT Tok. | 0.914 | 595.0 | 0.948 | 570.2 | **0.0047** |

[†]Results from survey paper by Stutz et al. (2018).
[‡]Full PASCALVOC12 due to folds for SBD missing from website.

The results in Table 3 suggests that interpretations extracted from the ViT and RViT models are less faithful to the predictions than interpretations procured with LIME with SLIC superpixels. Contrarily, the predictions extracted from the attention flow and PCA using the SPiT model provide *better comprehensiveness scores* than interpretations from LIME, indicating that SPiT models produce interpretations that more effectively exclude irrelevant regions of the image. A one-sided $t$-test confirms that the improvement in comprehensiveness between ATT.FLOW and LIME for the SPiT model is statistically significant.[1] Furthermore, we note that the sufficiency score for SPiT models are closer to the baseline LIME interpretations than what we observe for the ViT, indicating that the interpretations from SPiT model captures the most essential features better than a canonical ViT. Figs. 1, D.1, D.2, and D.3 clearly shows that the granularity of superpixel tokens provide interpretations that closely align with the semantic content of the image.

## 3.2 ABLATIONS

**Tokenizer Generalization**: In Section 2.1 we outlined our framework, and in Section 2.4 we showed that our framework generalizes the canonical ViT. This allows us to contrast the generalizability of the different strategies across models by directly swapping tokenization strategies between the models. We report the change in accuracy (Δ Acc.) of models when changing tokenizers in Table 4.

Unsurprisingly, the ViT with square tokenization performs relatively poorly when evaluated on irregular patches. We note that the RViT models trained with the random Voronoi tokenization see an increase in accuracy when evaluated over square patches. Furthermore, we see that the SPiT models also generalize well to both to square and Voronoi tokens, but is highly dependent on the gradient features. In particular, when gradient features are included, we see a very low drop in accuracy when evaluating over Voronoi tokens for the SPiT model, as well as evaluating superpixel tokens for the RViT models. This largely confirms our conjecture that gradient features help encode important information about texture, scale, and shape for irregular patches.

**Quality of Superpixels**: To evaluate the quality and estimate the information loss of our proposed superpixel segmentation, we compute the *explained variation* given by

$$\mathbb{V}_{\text{Expl.}}(\pi \mid \xi) = \frac{1}{\mathbb{V}(\xi)} \sum_{S \in \pi} \Pr(S) \big(\mathbb{E}(\xi \cap S) - \mathbb{E}(\xi)\big)^2, \tag{6}$$

where $\Pr(S) = |S|/|\xi|$. The explained variation quantifies how well the superpixels capture the inherent structures in an image by measuring the amount of dispersion in the image which can be attributed to the partitioning $\pi$. An ideal algorithm would produce a high $\mathbb{V}_{\text{Expl.}}$ with a minimal number of superpixels. We compare our approach with SotA superpixel methods (Stutz et al., 2018) in Table 5, demonstrating that our superpixel algorithm performs comparably to SotA algorithms with substantially lower inference time, which is crucial for on-line tokenization.

---

[1]One-sided $t$-test (ATT.FLOW > LIME): ($t = 6.54, p < 10^{-10}, \mathrm{df} = 49664$).

## 4 DISCUSSION AND RELATED WORK

**Related Work**: Recent research show that interest in adaptive tokenization is burgeoning in the field. We propose a taxonomy of adaptive tokenization with two primary dimensions for dividing the approaches—cf. Fig. 4 for an illustration of this spectrum. Firstly, the division is based on the *coupling or integration* of tokenization into the transformer architecture. Some approaches (Ma et al., 2023; Huang et al., 2022; Bolya et al., 2023) prioritize this coupling aspect. In contrast, others adopt a decoupled modular approach (Havtorn et al., 2023; Ronen et al., 2023), which aligns with our SPiT framework. Moreover, to enhance our understanding of this taxonomy, we introduce the second dimension of *token granularity*. This dimension enables us to assess the proximity of a method to operating with pixel-level precision. By considering both dimensions, we can better comprehend the full spectrum of adaptive tokenization for transformers.

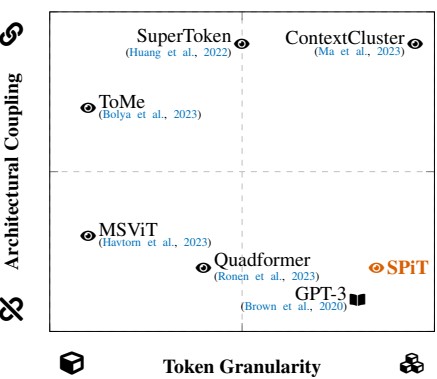

**Figure 4:** Taxonomy of adaptive tokenization in transformers. Tokenization ranges from decoupled (⊗) to coupled (🔗) to the transformer architecture, and from coarse (🎲) to fine (🧩) token granularity. To contextualize vision models (👁) with LLMs (📖), GPT-3 (Brown et al., 2020) is included for reference.

A significant body of current research is primarily designed to improve scaling and reducing overall compute for attention (Bolya et al., 2023; Ryoo et al., 2021; Yuan et al., 2021) by leveraging token merging strategies in the transformer layers with square patches, and can as such be considered *low-granularity coupled approaches*. Distinctively, the SuperToken transformer (Huang et al., 2022) applies a coupled approach to extract a non-uniform token representation. The approach is fundamentally patch based, and does not aim for pixel-level granularity.

In contrast, recent work on multi-scale tokenization (Havtorn et al., 2023; Ronen et al., 2023) have made strides towards a *decoupled approach* where the tokenizer is largely independent of the transformer architecture. These works are commensurable with *any transformer backbone*, including our SPiT framework, and can notably be used to improve computational overheads. While they operate on a *lower level of granularity* with square tokens, there is significant potential for synergy between these approaches and our own, particularly given the hierarchical nature of SPiT. On the periphery, Ma et al. (2023) propose a pixel-level kernelized approach in a *coupled high granularity* approach.

**Further Work**: Our work is distinguishable as a *decoupled high-granularity apprach* that creates multiple paths for further work. For more coherence in dense predictions, heuristic algorithms for on-line superpixel tokenization should be replaced with a learnable framework. Since our framework leverages a hierarchical graph, we see strong potential in exploring graph neural networks (GNNs) for tokenization, while the hierarchy can be directly applied with self-supervised frameworks such as DINO (Caron et al., 2021), or multiscale pyramid attention models (Wang et al., 2021; 2022a) in a coupled approach.

The modularity of our framework provides opportunites for research into the dynamic between ViTs and tokenization. Coupling SPiT with gated mixed-scale dynamical tokenization (Havtorn et al., 2023) could be applied to further improve scalability, and potentially be used for learnable tokenization. More work can be done in studying the effects of irregularity of partitions, as mentioned in Section 3.2, while random voronoi tesselations have opportunities for using stochastic tokenization as an augmentation strategy, which could be interesting for self-supervised frameworks.

**Conclusion:** In this work, we demonstrated that our proposed tokenization strategies generalize transformers for vision tasks, and that irregular patches can be successfully leveraged for training powerful models. We showed that SPiT models show strong performance in salient segmentation, increase faithfulness of the interpretablity of attributions, while producing comparable results to canonical ViT models for classification tasks. Our proposed gradient features and positional encoding improve performance of canonical ViTs with base capacities, and that irregular tokenizers generalize between different tokenization strategies.

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

# A    EQUIVALENCE OF FRAMEWORKS

**Definition A.1** (ViT Tokenization). Let $\xi \colon H \times W \to \mathbb{R}^C$ be an image signal with tensor representation $\vec{\xi} \in \mathbb{R}^{H \times W \times C}$. The canonical ViT tokenization operator $\tau^* \colon \mathbb{R}^{H \times W \times C} \to \mathbb{R}^{N \times \rho \times \rho \times C}$ partitions the image into $N = \lceil \frac{H}{\rho} \rceil \cdot \lceil \frac{W}{\rho} \rceil$ non-overlapping $C$-channel square zero-padded patches. For $H \bmod \rho = W \bmod \rho = 0$, we have that $N = \frac{H}{\rho} \cdot \frac{W}{\rho}$, and no padding is necessary.

**Definition A.2** (ViT Features). Let $\rho$ denote the patch dimension of a canonical ViT tokenizer $\tau^*$, and let $M = \rho^2 C$. The canonical ViT feature extractor $\phi^* \colon \mathbb{R}^{N \times \rho \times \rho \times C} \to \mathbb{R}^{N \times M}$ is given by $\phi^* = \mathrm{vec}_M$, where $\mathrm{vec}_M$ denotes the vectorization operator applied independently to each of the $N$ patches via $\rho \times \rho \times C \mapsto M$.

**Definition A.3** (ViT Embedder). Let $\phi^*$ be a canonical ViT feature extractor, and let $Q \in \mathbb{R}^{N \times D}$ denote a positional encoding. The canonical ViT embedder $\gamma^* \colon \mathbb{R}^{N \times M} \to \mathbb{R}^{N \times D}$ is given by

$$\gamma^*(z) = L_\theta z + Q$$

where $L_\theta \colon \mathbb{R}^{N \times M} \to \mathbb{R}^{N \times D}$ is a learnable linear transformation, and $Q$ is either a learnable set of parameters or a function of the positions of the $N$ blocks in the partitioning induced by the canonical tokenizer $\tau^*$.

**Lemma A.4** (Feature Equivalence). *Let $\tau^*$ denote a canonical ViT tokenizer with a fixed patch size $\rho$, and let $\phi$ denote a gradient excluding interpolating feature extractor with $\beta = \rho$. Then the operations $\phi \circ \tau^*$ are equivalent to the canonical ViT operations $\phi^* \circ \tau^*$.*

> **Proof.**    The proof is highly trivial but illustrative. Note that for each of the $N$ square patches generated by $\tau$, the extractor $\phi$ performs an interpolation to rescale the patch to a fixed resolution of $\beta \times \beta$. However, for $\beta = \rho$ the patches already match the target dimensions exactly. It follows that the interpolation operation reduces to identity. The vectorization operator is equivalent for both mappings, hence $\phi = \mathrm{vec}_N = \phi^*$.    $\square$

**Proposition 2.1** (Embedding Equivalence). *Let $\tau^*$ denote an canonical ViT tokenizer with a fixed patch size $\rho$, let $\phi$ denote a gradient excluding interpolated feature extractor, and let $\gamma^*, \gamma$ denote embedding layers with equivalent linear projections $L_\theta^* = L_\theta$. Let $\Xi^{(\mathrm{pos})} \in \mathbb{R}^{N \times \beta^2}$ denote a matrix of vectorized joint histogram positional embeddings under the partitioning induced by $\tau^*$. Then for $H = W = \beta^2 = \rho^2$, the embeddings given by $\gamma \circ \phi \circ \tau^*$ are equivalent to the canonical ViT embeddings given by $\gamma^* \circ \phi^* \circ \tau^*$ up to proportionality.*

> **Proof.**    We first note that we can assume $\Xi^{(\mathrm{pos})}$ is a matrix with single entry components, since under $\beta = \rho$ and $N = \beta^2$, each vectorized histogram feature is a scaled unit vector $c_n \vec{e}_n$ with $n = 1, \ldots, N$. Moreover, since the partitioning inferred by $\tau^*$ exhaustively covers the spatial dimensions $H \times W$, the histograms essentially span the standard basis, such that $\Xi^{(\mathrm{pos})}$ is diagonal. Furthermore, since each patch is of the same size we have equal contribution towards each entry, such that $c_n = c_m$ for all $m \neq n$. Therefore, without loss of generality, we can ignore the scalars and simply consider $\Xi^{(\mathrm{pos})} = I$ as an identity matrix. From Proposition A.4 we have that $z = (\phi^* \circ \tau^*)(\vec{\xi}) = (\phi \circ \tau^*)(\vec{\xi})$. Then, since
>
> $$\gamma^*(z) = L_\theta z + Q = [L_\theta, Q] \begin{bmatrix} z \\ I \end{bmatrix} = \gamma(z) \tag{7}$$
>
> we have that $\gamma = \gamma^*$ up to proportionality for some constant $c = c_n$.    $\square$

*Remark* A.5. While we only demonstrate the equality up to proportionality, this can generally be ignored since we can effectively choose our linear projection under $\gamma$ to be $L_\theta / c$. We note that while the equality holds for empirical histograms, equality does not strictly hold for $\Xi^{(\mathrm{pos})}$ computed using KDE with a Gaussian kernel, however we point out that the contribution from the tails of a kernel $K_\sigma$ with a small bandwidth is effectively negligible.

## B  PREPROCESSING AND SUPERPIXEL FEATURES

Compared to standard preprocessing, we use a modified normalization scheme for the features for improving the superpixel extraction. We apply a combined contrast adjustment and normalization function using a reparametrized version of the Kumaraswamy CDF. which is computationally efficient and allows more fine-grained control of the distribution of intensities than empirical normalization, which improves the superpixel partitioning.

The normalization uses a set of means $\mu$ shape parameters $\lambda$ for normalizing the image and adjusting the contrast. The normalization is given by

$$\left(1 - \left(1 - x^\lambda\right)^b\right),\tag{8}$$

where $b$ is defined by

$$b = -\frac{\ln(2)}{\ln\left(1 - \mu^\lambda\right)},\tag{9}$$

and we set means $\mu_r = 0.485, \mu_g = 0.456, \mu_b = 0.406$ and $\lambda_r = 0.539, \lambda_g = 0.507, \lambda_b = 0.404$, respectively. This gives a normalized image with support in $[-1, 1]$.

The features used for the superpixel extraction are further processed using anisotropic diffusion, which smoothes homogeneous regions while avoiding blurring of edges. This technique was advocated for superpixel segmentation by Xiaohan et al. (1992). We use the algorithm proposed by Perona & Malik (1990) over 4 iterations, with $\kappa = 0.1$ and $\gamma = 0.5$. Note that these features are only applied for constructing the superpixels in the tokenizer. We emphasize that we do not apply anisotropic diffusion for the features in the predictive model.

## C  TRAINING DETAILS

As mentioned in Section 1.2, we use standardized ViT architectures and generally follow the recommendations provided by Steiner et al. (2021). We provide training logs, pretrained models, and code for training models from scratch in our GitHub project repository.

**Classification**: Training is performed over 300 epochs using the ADAMW optimizer with a cosine annealing learning rate scheduler with 5 epochs of cosine annealed warmup from learning rate $\eta_{\text{start}} = 1 \times 10^{-5}$. The schedule maxima and minima are given by $\eta_{\text{max}} = 3 \times 10^{-3}$, and $\eta_{\text{min}} = 1 \times 10^{-6}$. We use a weight decay of $\lambda_{\text{dec}} = 2 \times 10^{-2}$ and set the smoothing term $\epsilon = 1 \times 10^{-7}$. In addition, we used stochastic depth dropout with a base probability of $p = 0.2$ in addition to the budget input dropout, limiting the number of seen tokens during training. Models were pretrained with spatial resolution $256 \times 256$.

For augmentations, we randomly select between using the RANDAUG framework at medium strength or using AUG3 framework by Touvron et al. (2022) including CUTMIX (Yun et al., 2019) with parameter $\alpha = 1.0$. We use RANDOMRESIZECROP using the standard scale $(0.08, 1.0)$ with stochastic interpolation modes. Since the number of partitions from the superpixel tokenizer are adapted on an image-to-image basis, we effectively regularize the maximum number of superpixels during training using a *budget dropout* to improve training times.

We found that a naive on-line computation of Voronoi tessellations was unnecessarily computationally expensive, hence we precompute sets of random Voronoi tessellations with 196, 256, and 576 partitions, corresponding to images of $224 \times 224$, $256 \times 256$, and $384 \times 384$ resolutions given patch size $\rho = 16$.

All training was performed on AMD MI250X GPUs. One important distinction is that we do not use quantization with `bfloat16` for training our models, instead opting for the higher 32-bit precision of `float32` since this improves consistency between vendor frameworks. Inference was carried out on a mixture of NVIDIA A100, RTX 2080Ti, Quadro P6000, and AMD MI250X to validate consistency across frameworks.

**Fine Tuning**: All base models were fine-tuned over 30 epochs with increased degrees of regularization. We increase the level of RANDAUG to "strong" using 2 operations with magnitude 20. Additionally, we increase the stochastic depth dropout to $p = 0.4$. Fine tuning was performed with

spatial resolution $384 \times 384$, and we reduce the maximum learning rate to $\eta_{\max} = 1 \times 10^{-4}$. For the alternative classification datasets CIFAR100 and CALTECH256, fine tuning was performed by replacing the classification head and fine tuning for 10 epochs using ADAMW with learning rate $\eta = 1 \times 10^{-4}$ and the same weight decay. No augmentation was used in this process, and images were re-scaled to $256 \times 256$ for training and evaluation.

## D  INTERPRETABILITY AND ATTENTION MAPS

For LIME explanations, we train a linear surrogate model $L_\Phi$ for predicting the output probabilities for the prediction of each model $\Phi$. To encourage independence between tokenizers and LIME explanations, as well as promote direct comparability, we use SLIC with a target of $|\pi| \approx 64$ superpixels. We use Monte Carlo sampling of binary features for indicating the presence or omission of each superpixel with stochastic $p \in \mathrm{Uniform}(0.1, 0.3)$, and keep these consistent across model evaluations. We observed that certain images in the IN1K at times produced less than 5 superpixels using SLIC, hence these images were dropped from the evaluation.

The attention flow (Abnar & Zuidema, 2020) of a transformer differs from the standard attention roll-out by accounting for the contributions of the residual connections in computations. The attention flow of an $L$-layer transformer is given by

$$A_{\mathrm{Flow}} = \prod_{i=1}^{L} \big( (1 - \lambda)I + \lambda A_i \big). \tag{10}$$

where we set $\lambda = 0.9$ to normalize the stochasticity while accentuating the contribution of the attention matrices. We use max-aggregation over the heads to extract a unified representation. Following Dosovitskiy et al. (2021) and Caron et al. (2021), we extract the attention for the class token as an interpretation of the model's prediction.

For the PCA projection, we take inspiration from the visualizations technique used in the work of Oquab et al. (2023). In this work, the features of multiple images with comparable attributes are concatenated, and projected onto a set of the top principal components of the image. We compute a set of 5 prototype centroids $\nu \in \mathbb{R}^{1000 \times d \times 5}$ for each class token of each model over ImageNet using KMeans, while enforcing relative subclass orthogonality by introducing a regularization term

$$J(\nu) = \frac{\lambda_\nu}{1000} \sum_{c=1}^{1000} \| I - \nu_c^\mathsf{T} \nu_c \|_2^2, \tag{11}$$

selecting $\lambda_\nu = 0.1$. Given a prediction $c$, we concatenate the prototypes to the token embeddings to form a matrix $M = [\Phi(\xi; \theta)^\mathsf{T}, \nu_c^\mathsf{T}]^\mathsf{T}$. Letting $U\Sigma V^\mathsf{T} = M - \mu(M)$ be a low-rank SVD of the centered features, we then project the original features to the principal components by $\Phi(\xi; \theta)V$, and use max-aggregation to extract the attribution as an interpretation of the model's prediction. We experimented with different ranks, but found that simply using the first principal component aligned well with attention maps and LIME coefficients. This somewhat mirrors the procedure by Oquab et al. (2023), where a thresholded projection on the first principal component is applied as a mask. In the interest of reproducibility, we provide links for downloading normalized attention maps for all attributions in our GitHub repository.

To quantify the faithfulness of the attributions for each model, we used comprehensiveness and sufficiency as proposed by DeYoung et al. (2020). Given a sequence of quantiles $Q \in [0, 1]$ from an attribution, these metrics are given by

$$\mathrm{COMP}_{Q|x,\Phi} = \frac{1}{|Q|} \sum_{q \in Q} \big( \Phi(x; \theta) - \Phi(x \setminus x_{>q}; \theta) \big), \tag{12}$$

$$\mathrm{SUFF}_{Q|x,\Phi} = \frac{1}{|Q|} \sum_{q \in Q} \big( \Phi(x; \theta) - \Phi(x \setminus x_{\leq q}; \theta) \big). \tag{13}$$

The benefit of these metrics is that they are symmetrical, and invariant to the scaling of the attributions due to applying quantiles to produce the masks. Following the procedure outlined by DeYoung et al. (2020) we set the quantiles to $Q = (0.01, 0.05, 0.2, 0.5)$.

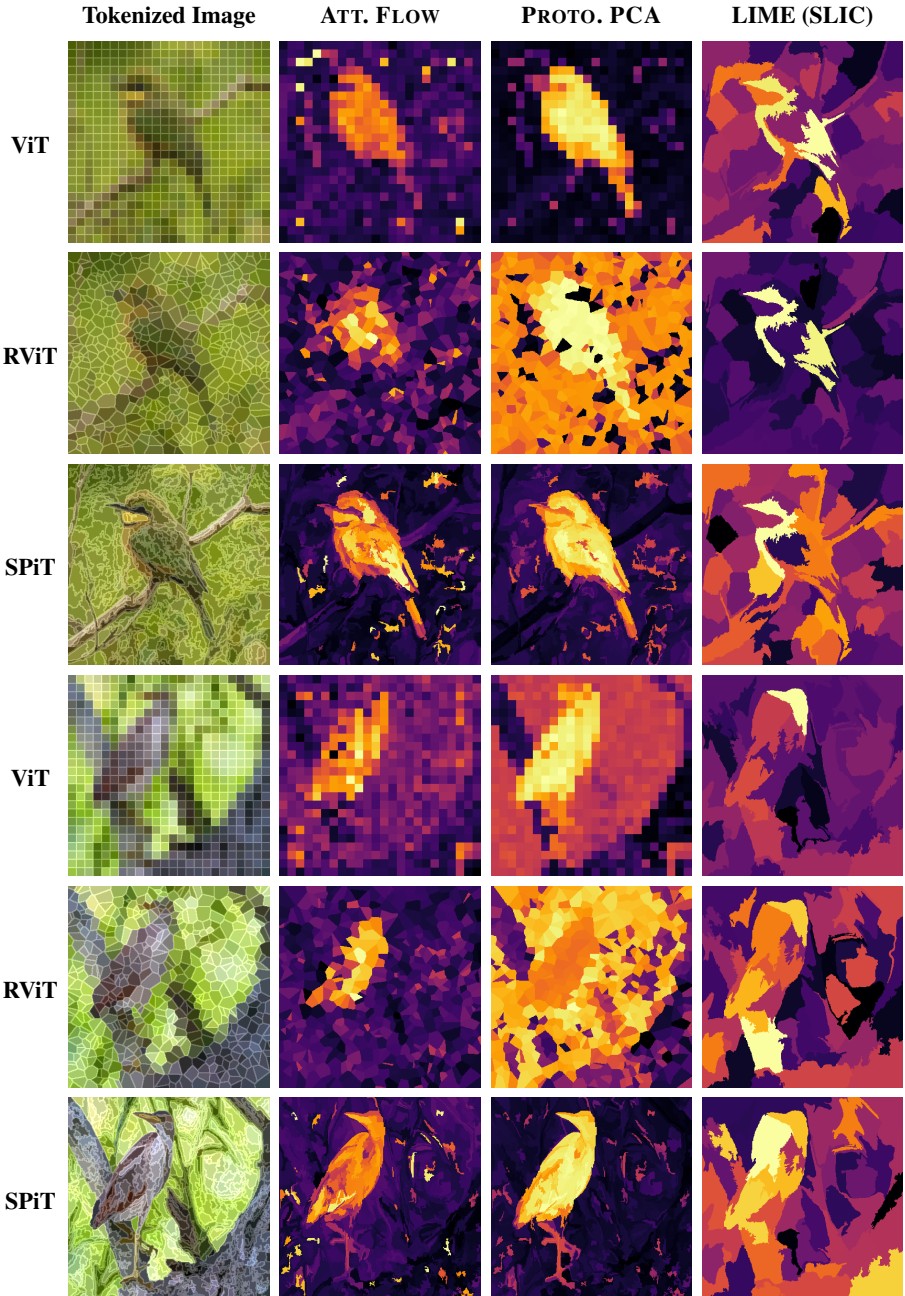

**Figure D.1:** Visualization of feature attributions for prediction "bee eater" and "bittern" with different tokenization strategies: square partitions (ViT), random Voronoi tesselation (RViT) and superpixels (SPiT).

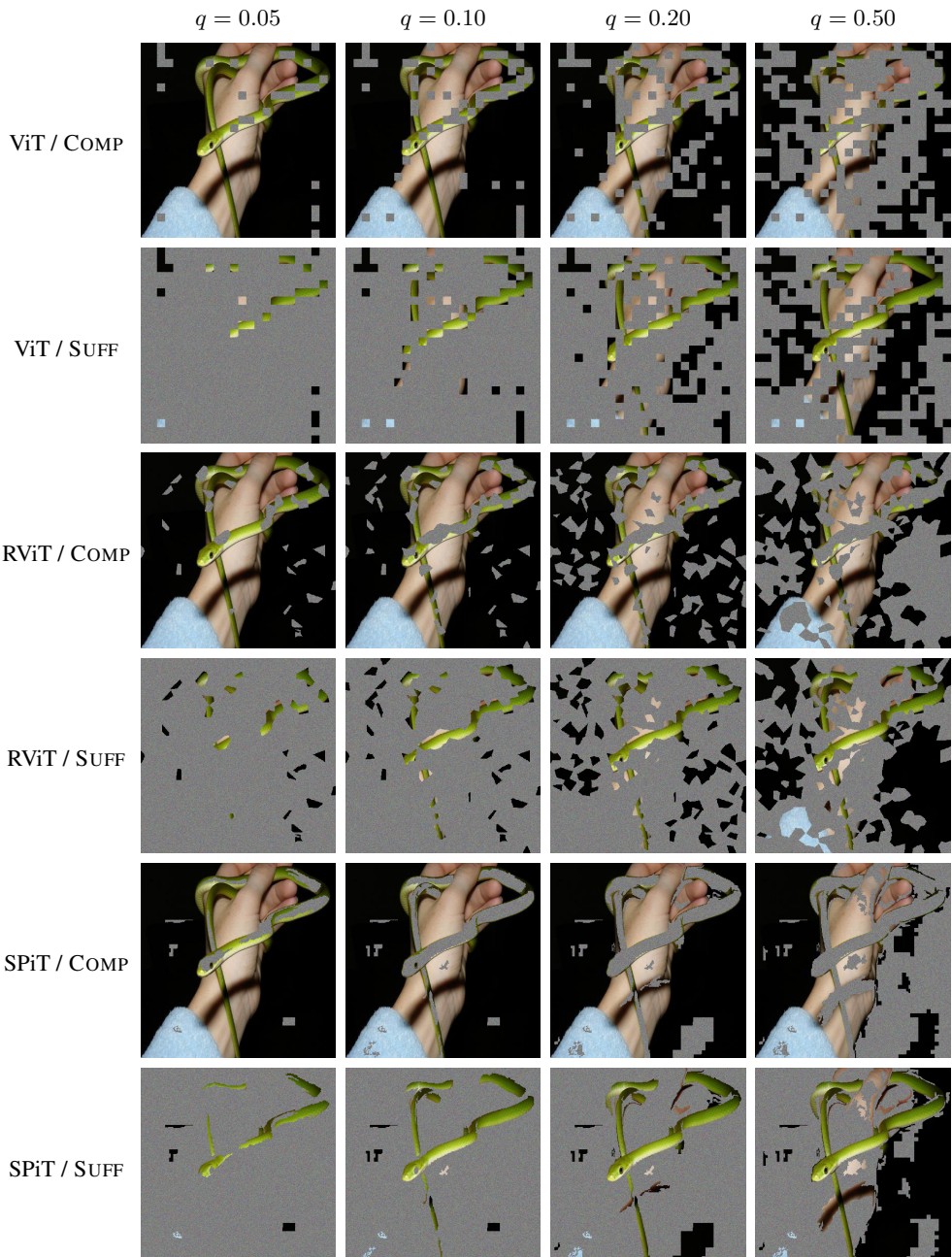

**Figure D.2:** Visualization of attention flow occlusions at different quantiles $q$ for prediction "grass snake". Note how the scaling of attention maps under superpixel tokenization improves occlusion for the predicted class.

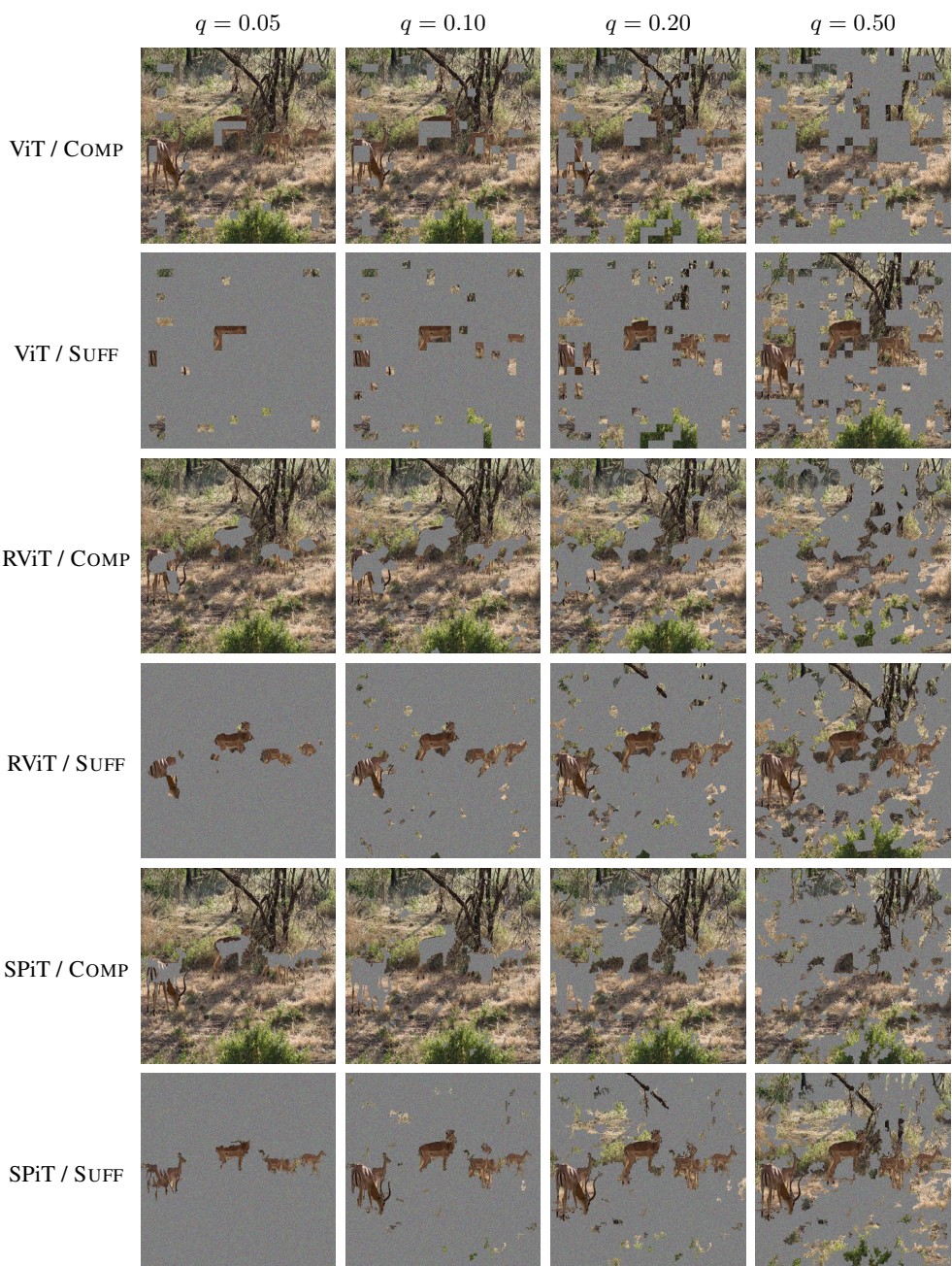

**Figure D.3:** Visualization of attention flow occlusions at different quantiles $q$ for prediction "impala". Note how the scaling of attention maps under superpixel tokenization improves occlusion for the predicted class.

**Table E.1:** Results w. CI (95%) for models with RVT tokenizers (5 runs).

| ViT Model | | | IN1K | INREAL | CIFAR100 | CALTECH256 |
|---|---|---|---|---|---|---|
| Name | Tok. Feat. Grad.Ft. | | Lin. | Lin. | Lin. | Lin. |
| SPiT-S16 RV | Intp. | ✗ | $0.7669 \pm 0.0002$ | $0.8285 \pm 0.0003$ | $0.8557 \pm 0.0028$ | $0.8521 \pm 0.0007$ |
| SPiT-S16 RV | Intp. | ✓ | $0.7593 \pm 0.0003$ | $0.8183 \pm 0.0002$ | $0.8563 \pm 0.0032$ | $0.8558 \pm 0.0006$ |
| SPiT-B16 RV | Intp. | ✗ | $0.7878 \pm 0.0002$ | $0.8436 \pm 0.0002$ | $0.8941 \pm 0.0043$ | $0.8731 \pm 0.0007$ |
| SPiT-B16 RV | Intp. | ✓ | $0.7892 \pm 0.0002$ | $0.8414 \pm 0.0001$ | $0.8875 \pm 0.0030$ | $0.8644 \pm 0.0006$ |

**Table F.1:** Estimated $\mathbb{E}(|\pi^{(T)}|)$ for SPiT tokenization over IN1K (training set, CI 95%).

| Im.Size | $\mathbb{E}(|\pi^{(1)}|)$ | $\mathbb{E}(|\pi^{(2)}|)$ | $\mathbb{E}(|\pi^{(3)}|)$ | $\mathbb{E}(|\pi^{(4)}|)$ |
|---|---|---|---|---|
| 224 | $11\,940.278 \pm 2.848$ | $3155.512 \pm 0.808$ | $794.650 \pm 0.209$ | $197.411 \pm 0.052$ |
| 256 | $15\,496.020 \pm 3.786$ | $4097.510 \pm 1.074$ | $1031.727 \pm 0.277$ | $256.051 \pm 0.071$ |
| 384 | $34\,084.297 \pm 9.188$ | $9047.289 \pm 2.586$ | $2287.822 \pm 0.669$ | $567.690 \pm 0.172$ |

# E  EXTENDED DISCUSSION ON RESULTS

Certain interesting observations can be made from our results in Table 1. Firstly, random Voronoi tessellations perform better than data-driven superpixels for gradient excluding features, and despite its inherent stochasticity, tokenization with random Voronoi tessellations proves to be a relatively effective strategy, and demonstrate surprisingly consistent results over prediction tasks as reported in Table E.1. To account for the stochasticity in validation, we compute accuracy scores over five runs and report 95% confidence intervals in Table E.1. We find that the segmentations based on the Voronoi tessellations produces remarkably consistent results over the validation set.

Additionally we note that gradient including tokenizers perform comparatively worse for small (S) models. This is particularly noteworthy, since the gradient features are essentially an added set of features to the model. We speculate that this could be an artifact of over-fitting on information-dense features, at the expense of the utility of the canonical pixel features.

# F  ADDITIONAL DETAILS ON SUPERPIXEL TOKENIZATION

**Number of Superpixels:** In section 2.2, we mention that SPiT gives comparable numbers of partitions to a ViT with different patch sizes. Table F.1 shows empirical results for superpixel sizes using the SPiT tokenizer over the training images of IMAGENET1K, and Fig. F.1 compares the results to number of patches with canonical ViT tokenization, demonstrating the validity of our claims.

Importantly, these results also reveal much about effective inference times. In Table 5, we show that the overhead for constructing the superpixels is very low. However, the number of tokens depends on the image. Images with large homogeneous regions will be processed faster, while images with many independent regions will necessary incur a cost. Nevertheless, the results in Table F.1 show

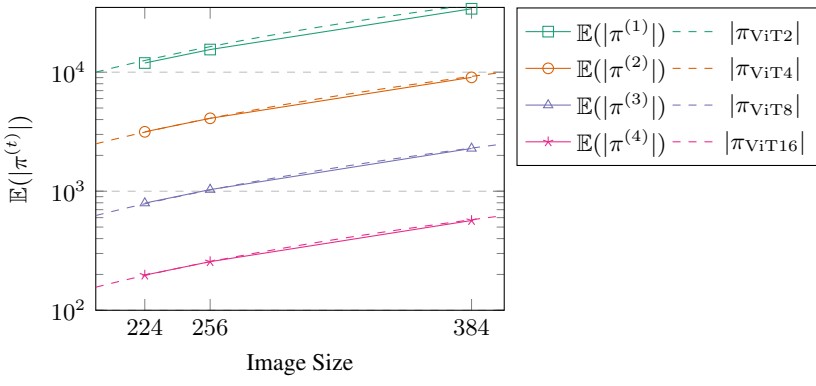

**Figure F.1:** Comparison $\mathbb{E}(|\pi^{(T)}|)$ for SPiT with different ViT patch sizes.

that we will, on average, have comparable inference times to a canonical ViT due to the beneficial properties of our proposed superpixel tokenization.

**Superpixel Compactness:** In the interest of reproducibility, we also outline a few additional details on our proposed superpixel tokenization. Readers familiar with classical superpixel algorithms may have noted that our exposition in Section 2.2 does not explicitly mention any regularization for compactness, which is seen as an attractive feature of a superpixel partition. In fact, our proposed tokenizer does support optional regularization of compactness based on a simplified measure of bounding box density. We outline the details of the regularization in the following paragraphs.

Recall that the size-based similarity for vertices $u, v$ when $u = v$ at step $t$ is computed by

$$w_\xi(u, v) = \frac{|\pi_u|^{(t)} - \mu_{|\pi|}^{(t)}}{\sigma_{|\pi|}^{(t)}}, \tag{14}$$

where $\mu_{|\pi|}^{(t)}$ is the mean size of partitions and $\sigma_{|\pi|}^{(t)}$ is the standard deviation. To control the effect of the regularization, this weight is then truncated to the range $[-\lambda_{\text{size}}, \lambda_{\text{size}}]$ where $\lambda_{\text{size}}$ is a hyperparameter. We set $\lambda_{\text{size}} = 0.75$ in our implementation.

Compactness is regulated through bounding box density $\delta_{\text{bbox}}(u, v)$ given by

$$\delta_{\text{bbox}}(u, v) = \frac{4(|\pi_u|^{(t)} + |\pi_v|^{(t)})}{\text{per}(u, v)^2}, \tag{15}$$

where

$$\text{per}(u, v) = \max_y(\pi_u, \pi_v) - \min_y(\pi_u, \pi_v) + \max_x(\pi_u, \pi_v) - \min_x(\pi_u, \pi_v) \tag{16}$$

corresponds to the perimeter of the bounding box that encapsulates superpixel vertices $u$ and $v$. The final similarity metric combines the weight function and the compactness regularization via

$$\lambda_{\text{bbox}}\delta_{\text{bbox}} + (1 - \lambda_{\text{bbox}}) \left( \frac{w_\xi(u, v) + 1}{2} \right), \tag{17}$$

where $\lambda_{\text{bbox}}$ serves as the hyperparameter for controlling the amount of compactness regularization. This emphasizes how tightly the two neighbouring superpixels $u$ and $v$ are packed in their bounding box. Higher values of $\delta_{\text{bbox}}$ mean the merging of the two vertices are spatially denser, forming a more compact structure. We recommend setting $\lambda_{\text{bbox}} = 0.1$ for SPiT models using gradient excluding feature extractors.

## G  UNSUPERVISED SALIENT SEGMENTATION

The TokenCut (Wang et al., 2022b) framework proposes to use a normalized cut (Shi & Malik, 2000) over the key features without class tokens in the last self-attention layer of the network. A soft adjacency $A_{\text{TC}}$ is computed using cosine similarities, which are thresholded using a small threshold $\tau_{\text{TC}} = 1/3$ to estimate adjacency over the complete graph over token features. The normalized cut is performed by extracting the Fiedler vector; the second smallest eigenvector of the graph Laplacian, and gives a bipartition of the graph into foreground and background elements. The original paper (Wang et al., 2022b) uses DINO (Caron et al., 2021) as a pretrained base model.

We found that extracting the key tokens from the last self-attention operator in the network is less effective than simply using the final features for the SPiT framework. In TokenCut, the saliency map is refined using postprocessing with a bilateral solver, however, in the SPiT framework this step is clearly redundant. Instead, we simply standardize the Fiedler vector using its mean and standard deviation, and map the result on the segmentations from the SPiT tokenizer. For certain images, the foreground and background elements could be swapped under the standard unsupervised normalized cut method. From our experiments on interpretability, we found that simply taking the class token for the full image, and comparing it using cosine similarity to class tokens (produced given the saliency mask) will accurately provide a robust estimate of which element is the foreground and the background.

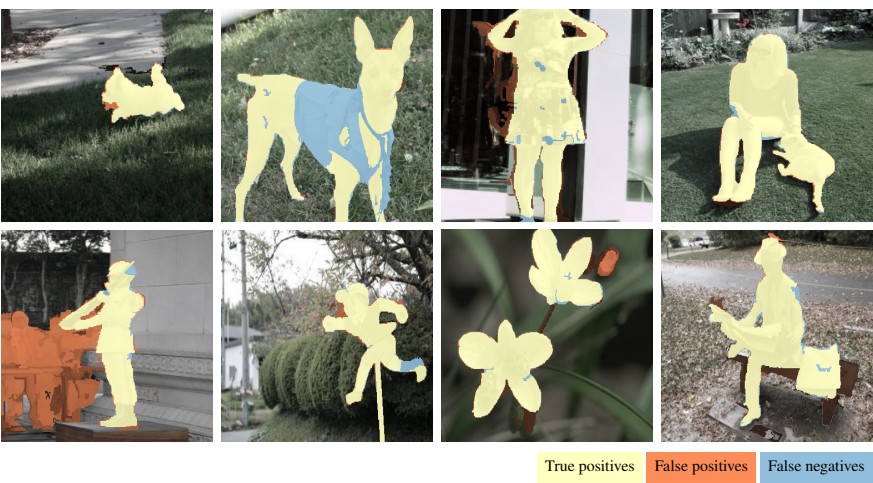

True positives    False positives    False negatives

**Figure G.1:** Non-cherry picked samples ({`0257..0264`}.`jpg`) of salient segmentation results on ECSSD.

