# OpenReview forum: "A Spitting Image: Superpixel Transformers"
_ICLR.cc/2024/Conference — Submitted to ICLR 2024_

### Official Review · Reviewer_Un4y · 2023-10-30

**Soundness:** 3 good
**Presentation:** 3 good
**Contribution:** 2 fair
**Rating:** 6
**Confidence:** 4

**Summary:**

The paper proposes to revisit the standard uniform square tokenization in vision transformers, and instead replace it with **superpixel-based tokenization**. The superpixel partition follows the methodology of "SuperPixel Hierarchy" (Wei et al, Transactions on Image Processing 2018) with an additional regularization term for self-loop edges. Based on this criterion, pixels are grouped hierarchically to form superpixels, and the number of hierarchy levels is chosen to obtain a comparable number of tokens to `ViT-B/16`.
Once the superpixels are defined, several sets of features are extract for each of them to form the input token representations:
  * **positional encodings** are computed as a histogram over the spatial positions present in the superpixel
  * **texture features** are extracted using gradient operators
  * **color features** are computed by interpolating the light intensity information present in the suerpixel

Once the superpixel and corresponding features are defined, the rest of the model is defined as a standard transformer architectures. In practice, the paper uses **ViT (Small and Base)** as backbone. The proposed method is evaluated on the task of image classification, as well as through various explainability metrics. It is compared to the backbone (ViT), as well to a method employing random Voronoi cells as tokens. The results show that the proposed method can match classification accuracy of the standard ViT backbone while yielding more interpretable feature attribution maps.

**Strengths:**

- **Clear writing and motivation**: The paper is clearly written and fully describes the proposed tokenization as well as feature extraction methods. The core motivation of departing from "uniform square patches" tokenization is very meaningful and superpixels appear like a well established tool to tackle this problem.

- **Clear reproducibility details**: The paper clearly describes the training details and hyperparameters in the appendix to reproduce the results.

- **Interpretability results**: The paper report interpretability experiments results using multiple metrics/frameworks to evaluate the faithfulness of feature attributions.

**Weaknesses:**

- **Practicality of superpixels**: One advantage of square tokenization is that it is highly practical:, it's a simple patching operation, and the position encodings can easily be transferred across scales using interpolation (see for instance "*FlexiViT: One Model for All Patch Sizes, Beyer et al, CVPR 2023*"). In contrast, the proposed method requires the extra step of generating superpixels for each input image, which is not discussed in the paper

- **Limited baselines:** The paper primarly evaluates three methods: `ViT-{S,B}/16` (the base model), `SPiT-{S, B}/16` (the proposed model) and `RViT-{S,B}/16`, which, from my understanding, is a baseline introduced in this paper using random Voronoi cells as tokens. I understand that improving classification accuracy is not the main goal of the paper hence the choice of baselines, but I do still think the paper would benefit from comparing to baselines which have a similar goal (i.e. either methods that propose more flexible tokenization, or more interpretable attention maps)

- **Missing related work:** In general, the discussion of related work in the paper is very short and does not seem to contextualize the paper enough. For instance, some points which may be interesting to discuss:
  * On the aspect of interpretability, models such as `DINO` trained with finer-grained tokens (*Emerging Properties in Self-Supervised Vision Transformers, Caron et al, CVPR 2021*) would be a stronger baseline than ViT trained with coarser patch size
  * *Vision Transformer with Super Token Sampling, Huang et al, CVPR 2023* is another method that uses superpixel in Vision transformers. The related work section explains that this method build super-pixel by gradually merging square patches, unlike the current work which uses super-pixels from the start; However, it is not clear to me why this is preferable, and how these two approaches compare in practice.
  * In addition, there has been many works investigating more flexible tokenization than the standard ViT "uniform square" assumption, which is not reflected in the introduction. For instance:
    * **Token merging** approaches (e.g. "*`ToME`: Your ViT but faster, Bolya et al, ICLR 2023*") also alleviate the issues that [quote] *" complexity and memory scales quadratically with the number of tokens in self-attention"*
    * Other works on **dynamic tokenization** (e.g. "*Vision Transformers with Mixed-Resolution Tokenization, Ronen et al, 2023*" or "*MSViT: Dynamic Mixed-Scale Tokenization for Vision Transformers, Havtorn et al, 2023*") also propose to incorporate mixed-resolution information directly at the input tokens stage and also show that ViT "can be succesfully trained under irregular tokenization"
    * `Swin` transformers and other **multi-scale works** gradually build multi-scale tokens, hence they do not have the issue that [quote] *"the scale of the partitions are rigidly linked to the model architecture by a fixed patch size"*

**Overall summary:** My main concern is that it is not clear to me why using superpixels as tokens is advantageous with respect to other methods implementing some form of input-dependent tokenization, either in terms of performance or computational cost. The main gain seems to be in obtaining more fine-grained interpretability maps, but these are also only compared to the standard ViT-backbone (and the random voronoi cell baseline)

**Post-rebuttal note:**  As some other reviewers mentioned, Is still have some reserves about the novelty/impact of the proposed method; however, in light of the rebuttal addressing several concerns and showing promising results on dense prediction tasks, I am raising my score from 5 to 6

**Questions:**

- **Notion of equivalence in Propositions 1 and 2**: What does it mean for the two operators to be equivalent ? the proof of proposition 1 only seems to show that both operators have the same input and output dimensions. Generally, I am not sure what the conclusion of Section 2.4 is and how significant it is.

- **Number of tokens and batched executions:** In Section 2.2, it is said that "*We empirically verify that setting T = 4 produces comparable numbers of tokens to a ViT-B16*"; Does this mean that each image has a different number of tokens in practice, and how does this impact batched execution during training/inference ?


- **Evaluation on segmentation:** The choice of super-pixels seem particularly relevant for more dense tasks such as segmentation: This might be a more favorable task to evaluate on than image classification.

---

> ### Author Response · Authors · 2023-11-17
>
> First and foremost, we emphatically praise the reviewer in their dedication to extensive constructive feedback to our work. We will do our best to address their concerns and clarify important details.
>
> > Practicality of superpixels: One advantage of square tokenization is that it is highly practical:, it's a simple patching operation, and the position encodings can easily be transferred across scales using interpolation (see for instance "FlexiViT: One Model for All Patch Sizes, Beyer et al, CVPR 2023").
>
> We understand the concerns the reviewer has with regard to the efficacy in construction of superpixels, however our proposed framework is neither at odds with, nor any form of replacement for FlexiViT. One of **our main contributions is our proposed extension to alternative tokenization strategies**, notably by **adapting ViTs to irregular partitions better aligned with semantic content in the image**. While we extract superpixels for each individual image, we emphasize that the superpixel extraction method is highly parallelizable, and is capable of extracting superpixels for thousands of images at once. Our tokenization and feature extraction methods are commensurable with canonical ViT architectures, as well as FlexiViT, and we extend ViTs to show that irregular non-convex patches or superpixels can be processed effectively with a general transformer architecture. Our approach has practical value both for extracting interpretations or dense predictions, since the patches are already aligned and contextualized within the image without requiring expensive post-processing to extract high resolution features, and note that several works have been proposed to tackle this issue (e.g. Amir et al. 2021, “Deep ViT features as dense visual descriptors”).
>
> > In contrast, the proposed method requires the extra step of generating superpixels for each input image, which is not discussed in the paper.
>
> Assuming that the reviewer refers to the overhead of computing the superpixels, we discuss this in Section 2.2 of our original submission. However, we see that this might not have been sufficiently clear in our original submission. Briefly summarized, the superpixels are generated using a graph approach, where each pixel is aggregated with its neighbor based on weighted similarity. In our revised submission, we show that this gives nearly exactly the same number of tokens as a canonical ViT with different patch sizes in the revised Table F.1 and Figure F.1. On average, $T=4$ produces nearly the exact same number of tokens in a ViT with patch size 16, while $T=3$ will obtain tokens equivalent on average to patch size 8. In light of the results in orig. Table 4 (rev. Table 5) the implications is that the effective increase in inference time (0.0047 sec./img.) will also be very close to the reported runtime of the SPiT tokenization step. As for overhead in training, we include throughput analysis in rev. Table 1. We include the results of Table F.1 here for convenience.
>
> | Im.Size | $\mathbb E(\lvert \pi^{(1)} \rvert)$ | $\mathbb E(\lvert \pi^{(2)} \rvert)$ | $\mathbb E(\lvert \pi^{(3)} \rvert)$ | $\mathbb E(\lvert \pi^{(4)} \rvert)$ |
> |---------|------------------------------------|------------------------------------|------------------------------------|------------------------------------|
> | 224     | 11940.278(2.848)                   | 3155.512(0.808)                    | 794.650(0.209)                     | 197.411(0.052)                     |
> | 256     | 15496.020(3.786)                   | 4097.510(1.074)                    | 1031.727(0.277)                    | 256.051(0.071)                     |
> | 384     | 34084.297(9.188)                   | 9047.289(2.586)                    | 2287.822(0.669)                    | 567.69(0.172)                      |

---

> ### Author Response · Authors · 2023-11-17
>
> > Limited baselines: The paper primarily evaluates three methods: ViT-{S,B}/16 (the base model), SPiT-{S, B}/16 (the proposed model) and RViT-{S,B}/16, which, from my understanding, is a baseline introduced in this paper using random Voronoi cells as tokens. I understand that improving classification accuracy is not the main goal of the paper hence the choice of baselines, but I do still think the paper would benefit from comparing to baselines which have a similar goal (i.e. either methods that propose more flexible tokenization, or more interpretable attention maps)
>
> > On the aspect of interpretability, models such as DINO trained with finer-grained tokens (Emerging Properties in Self-Supervised Vision Transformers, Caron et al, CVPR 2021) would be a stronger baseline than ViT trained with coarser patch size
>
> We agree with the reviewer that one must have a comparison with methods with similar goals to establish fair comparisons. While the reviewer points out several interesting works, they are geared towards scalability and efficiency in reducing the number of tokens, while our approach targets semantically aligned patches with pixel level granularity. Under this light, comparing against other methods doing tokenization with the goal of efficiency or other training paradigms; e.g., DINO (Caron et al. 2021), will not constitute a fair comparison. On the other hand, we agree that a more general evaluation of the method could provide insights about the separation of the tokenizer and the transformer backbone.
> To address the concern, we establish a taxonomy of adaptive methods in our related work section, and include another downstream task evaluation for unsupervised salient semantic segmentation, which compares our method to DINO to demonstrate the robustness of our proposal.
>
> > Missing related work: In general, the discussion of related work in the paper is very short and does not seem to contextualize the paper enough.
>
> > In addition, there has been many works investigating more flexible tokenization than the standard ViT "uniform square" assumption, which is not reflected in the introduction.
>
> We thank the reviewer for the suggestions about the literature that we missed before.  In the revised manuscript, we added the suggested references as well as the others from the other reviewers.
>
> We have significantly revamped the related work section thanks to the suggestions given by all the reviewers.  In the revised version, we touch upon the growing literature on tokenization in the transformers architectures and how **they have been focusing on changing the architectures to improve their efficiency**.  On the contrary, **we emphasize that we are working towards the complementary goal of decoupling the architectures** and bringing to the spotlight the vital role of the forgotten tokens. We outline a taxonomy for comparing these methods, highlighting potential benefits with the different approaches.
>
> > Token merging approaches (e.g. "ToME: Your ViT but faster, Bolya et al, ICLR 2023") also alleviate the issues that [quote] " complexity and memory scales quadratically with the number of tokens in self-attention"
>
> ToMe (Bolya et al. 2023) represents an adaptive method for effectively reducing the number of tokens between transformer blocks, which has strong merit in continued research in ViT architectures. Our work is distinct by the fact that we decouple the tokenization from the general architecture, with pixel-level granularity in the tokens. Our method is commensurable with ToMe, but the goal is to extract high resolution feature maps, and align tokens with semantic content in the image as opposed to improving efficiency.
>
> > Swin transformers and other multi-scale works gradually build multi-scale tokens, hence they do not have the issue that [quote] "the scale of the partitions are rigidly linked to the model architecture by a fixed patch size"
>
> Swin transformers approach the issues of quantisation issues with square partitions by grouping finer grain tokens differently in each step to alleviate some of the quantization artifacts in the ViT tokenization. This is somewhat similar to Tokens2Token (Yuan et al. 2021), but more computationally expensive. In our framework, we instead look to align the tokens with the semantic content directly, and decouple the tokenization from the transformer backbone.

---

> ### Author Response · Authors · 2023-11-17
>
> > Vision Transformer with Super Token Sampling, Huang et al, CVPR 2023 is another method that uses superpixel in Vision transformers. The related work section explains that this method build super-pixel by gradually merging square patches, unlike the current work which uses super-pixels from the start; However, it is not clear to me why this is preferable, and how these two approaches compare in practice.
>
> The reviewer touches on several relevant points with this observation, and is correct in pointing out that our proposed approach decouples tokenization from the backbone (as in LLMs) whereas SuperTokens apply tokenization coupled with the transformer backbone. Our original submission noted that SuperToken Sampling (Huang et al. 2023) is not operating with pixel level granularity, but instead uses a minimal discrete unit of a square patch. While we do not claim that our approach is superior to coupled tokenization, we claim there is inherent benefit in continued research towards both goals. We extend our discussion on these paradigms in Section 3.4 and 4 to emphasize our argument, and thank the reviewer for their observation.
>
> In principle, we argue that tokenization in LLMs is independent but instrumental in their efficacy, and by keeping the tokenizer modular, we significantly improve research towards adaptable transferable models (see our discussion in Table 3 / rev. Table 4). In light of the reviewers comment, it becomes clear that our original submission does not sufficiently address some of the key benefits that modular tokenization can provide. Firstly, a modular tokenizer could perceivably be optimized in an expectation-maximization scheme for any given ViT model. Secondly, it is reasonable to assume that different tasks could require different tokenization strategies. If tokenization schemes are embedded in the transformer blocks, one concern is that this could potentially negatively affect transferability, and new models might have to be retrained for each task.
>
> > Other works on dynamic tokenization (e.g. "Vision Transformers with Mixed-Resolution Tokenization, Ronen et al, 2023" or "MSViT: Dynamic Mixed-Scale Tokenization for Vision Transformers, Havtorn et al, 2023") also propose to incorporate mixed-resolution information directly at the input tokens stage and also show that ViT "can be succesfully trained under irregular tokenization"
>
> > My main concern is that it is not clear to me why using superpixels as tokens is advantageous with respect to other methods implementing some form of input-dependent tokenization, either in terms of performance or computational cost. The main gain seems to be in obtaining more fine-grained interpretability maps, but these are also only compared to the standard ViT-backbone (and the random voronoi cell baseline)
>
> We kindly thank the reviewer for bringing the pioneering work by Ronen et al. (2023) and Havtorn et al. (2023) to our attention, as mixed-scale tokenization ultimately shares similar goals as our own. We emphatically agree that these methods are highly interesting, but note that these have their own distinct focus and objective. Mixed-scale tokenization approaches focus on more clever partitioning strategies with the goal of improving efficiency, scaling, and complexity of transformer models. Our approach, as noted by the reviewer, is to construct data driven partitions independent of the general transformer backbone that semantically align with the image content. In our opinion, **the key difference** that distinguishes our work from the Quadtree approach (Ronen et al.) as well as the gating mechanism in MSViT (Havtorn et al.) is that **we use pixel-level granularity with irregular (non-convex, non-square) partitions that fully adapt to the semantic content**. We see inherent benefits in different tokenization strategies for different problems, and note that given their modularity, these works instead contextualize and augment our own contributions, and demonstrate the importance of research into novel tokenization approaches.

---

> ### Author Response · Authors · 2023-11-17
>
> > Notion of equivalence in Propositions 1 and 2: What does it mean for the two operators to be equivalent ? the proof of proposition 1 only seems to show that both operators have the same input and output dimensions. Generally, I am not sure what the conclusion of Section 2.4 is and how significant it is.
>
> The reviewer addresses the perceived trivial nature of Proposition 1. In the provided proof, we ourselves comment on this while pointing out that it is necessarily instructive. *It anchors interpolating feature extractors to canonical ViT features since they are equivalent given similar patch sizes.* Proposition 1 is formulated in the context of Proposition 2, where we show that our proposed method for positional embeddings (which is designed for irregular patches) exactly coincide with learnable positional embedding in ViTs. This posits all models in the same framework; ViTs uses fixed square “superpixels”, RViT uses stochastic Voronoi “superpixels”, and SPiT uses our proposed superpixel tokenizer, but all are trained with the same framework. **The purpose is to clearly show the commensurability between our framework and feature extraction in canonical ViTs.** In light of the reviewer's comment, we relabel the proposition as a lemma.
>
> > Number of tokens and batched executions: In Section 2.2, it is said that "We empirically verify that setting T = 4 produces comparable numbers of tokens to a ViT-B16";
>
> The reviewer congruously brings attention to a clear gap in our original submission. These are clearly experimental results that are of interest to readers in our field. **Full experimental results on the validity of this claim have been included in Table F.1 in our revised submission.**
>
> > Does this mean that each image has a different number of tokens in practice, and how does this impact batched execution during training/inference ?
>
> Exactly right. In our work, we discuss how we can regularize the number of superpixels, but not exactly control it, similar to nearly all other superpixel approaches. In our framework, all images are processed as a large supergraph consisting of images as a smaller subgraph, similar to the approach in GNNs. This allows us to handle variable token sizes effectively. To apply optimized computations in self attention frameworks (e.g. FlashAttention), we apply similar techniques as language transformers using masks. We realize that this point is not immediately clear in our original submission, and have revised Appendix C with more details to address this issue.

---

> ### Author Response · Authors · 2023-11-17
>
> > Evaluation on segmentation: The choice of super-pixels seem particularly relevant for more dense tasks such as segmentation: This might be a more favorable task to evaluate on than image classification.
>
> While we acknowledge that segmentation is well aligned with superpixel tokenization, we note that classification models (alongside contrastive learning paradigms) are often considered more or less fundamental. Very few, if any practitioners, leverage segmentation models for classification. On the other hand, starting with pre-trained classification models for segmentation is not only common, but generally advisable. Furthermore, we had a scientific curiosity in determining the properties of different tokenization strategies in ViTs, in particular in relation to graph based methods. Given comparable results on classification tasks, improved interpretability and explainability, and our results on generalizability of tokenization across models, we considered our work as both interesting and novel enough for publication.
>
> However, we need to extend our sincerest gratitude to the reviewer for suggesting applications with segmentation tasks, since they prompted us to reevaluate how we could demonstrate feasibility for dense prediction tasks within the current framework. In considering zero-shot and unsupervised segmentation approaches, we decided that unsupervised salient segmentation could be well aligned with this goal, and we revised our submission to include experiments with this task in rev. Table 2 and Appendix G. **We happily report SotA results on three datasets for unsupervised salient segmentation tasks**, and report our results here for convenience.
>
> | TokenCut Backbone | Postproc. | $\max F_\beta$ (ECSSD) | IoU (ECSSD) | Acc. (ECSSD) | $\max F_\beta$ (DUTS) | IoU (DUTS) | Acc. (DUTS) | $\max F_\beta$ (DUT-OMRON) | IoU (DUT-OMRON) | Acc. (DUT-OMRON) |
> |-------------------|-----------|-----------------------|-------------|--------------|----------------------|------------|-------------|--------------------------|-----------------|------------------|
> | DINO   | ❌        | 0.803                 | 0.712       | 0.918        | 0.672                | 0.576      | 0.903       | 0.600                    | 0.533           | 0.880            |
> |                   | ✔️        | 0.874                 | 0.772       | **0.934**    | 0.755                | 0.624      | **0.914**   | 0.697                    | **0.618**       | **0.897**        |
> | SPiT              | ❌        | **0.903**             | **0.773**   | **0.934**    | **0.771**            | **0.639**  | 0.894       | **0.711**                | 0.564           | 0.868            |

---

> ### Comment · Reviewer_Un4y · 2023-11-21
> **Thanks for your response**
>
> Dear authors,
> Thanks a lot for the detailed response!
> - It's nice to see that superpixel tokenizations brings an added benefit for dense prediction tasks
> - Thanks for adding results on the number of tokens generated by SPiT
> - Thanks for adding a more in-depth discussion of related work
>
> As some other reviewers mentioned, Is still have some reserves about the novelty/impact of the proposed method; however, in light of the rebuttal addressing several concerns and showing promising results on dense prediction tasks, I am raising my score to 6

---

> > ### Author Response · Authors · 2023-11-22
> > **Thank you for constructive insights**
> >
> > Thank you for your detailed and constructive feedback, and for acknowledging the strengths in our revised manuscript. Your comments have been instrumental in improving our work.
> >
> > We note your concerns regarding the novelty and impact of our method, and while we strived to address these aspects, we are intrigued by your perspective. Could you please elaborate on your specific reservations in this regard? Understanding your viewpoint will be beneficial for our continued research. We respect your expert opinion and look forward to any additional insights you might offer.

---

> > > ### Comment · Reviewer_Un4y · 2023-11-23
> > > **Response to authors**
> > >
> > > Dear authors,
> > > yes, of coursel to be more specific, my point on novelty also takes into account other reviews' assessment on topics I'm less of an expert in (e.g. superpixel extraction), and I may further increase my opinion on this after the discussion period. But If I summarize what I currently see as the main contributions of the paper and how they relate to existing literature, I would highlight the following:
> > >
> > > - **Method**
> > >   - **Showing that we can decouple tokenization from the transformer:** I think it's an interesting motivation but was also addressed in previous literature (for instance from the original ViT paper, where one can change the tokenization grid/resolution at inference by interpolating the positional embeddings, or from the mixed-scale tokenization line of work)
> > >   - **using super-pixels as tokens**: Some previous works have also considered using similar superpixel extraction pipeline with ViT (e.g. mentioned by reviewer Apxd)
> > >
> > > - **Experiments**
> > >   - **classification results:** using super-pixel tokenization performs on-par with standard ViT baselines
> > >   - **dense prediction results:** using super-pixel tokenization outperforms standard tokenization, which is a promising angle
> > >   - **interpretability:** using super pixels tokenization also seems to yield better saliency maps than standard tokens for roughly the same number of tokens.

---

### Official Review · Reviewer_N399 · 2023-11-01

**Soundness:** 3 good
**Presentation:** 3 good
**Contribution:** 3 good
**Rating:** 8
**Confidence:** 4

**Summary:**

This paper introduces an innovative approach to improving tokenization strategies in vision transformers. In contrast to traditional regular grid partitioning methods, they propose a superpixel transformer approach capable of irregular tokenization. By employing state-of-the-art metrics, this method can flexibly divide irregular patches, thereby enhancing the ability of subsequent transformer modules to extract higher-quality features and significantly improving model interpretability. Furthermore, the authors conducted an extensive series of experiments to rigorously validate the effectiveness of their method.

**Strengths:**

This paper demonstrates a highly promising starting point by introducing the use of superpixel partitioning for irregular patch division, which seeks to address the inherent limitations of traditional grid-based partitioning methods in transformers. As highlighted by the authors, this approach has the potential to significantly enhance the interpretability of attention mechanisms in Vision Transformer (ViT) and facilitate the extraction of high-quality features, marking its potential impact on pioneering applications of transformers. The paper maintains a clear and informative structure. In summary, this work demonstrates a high degree of originality and holds substantial potential for further research and applications in the field.

**Weaknesses:**

1、In the experimental section, the authors have provided thorough validation and analysis of the enhancement in feature extraction achieved by superpixel tokens in subsequent transformer modules. However, the paper lacks an analysis of SPiT efficiency. While Table 4 indicates that the superpixel algorithm used in this study performs comparably to state-of-the-art algorithms with substantially lower inference time, it is essential to explore whether the integration of superpixel tokens with subsequent transformer modules introduces a significant computational load. To address this, the authors should consider adding an additional set of comparative experiments to analyze the inference efficiency between ViT, RViT, and SPiT.
2、The determination of hierarchical levels T is not clearly explained. To provide a more comprehensive understanding, the authors could include an additional set of ablation experiments to explore the implicit relationship between T settings and model performance. These additional experiments and analyses will further strengthen the paper's contributions and insights.
3、The paper's summary of related work is relatively limited. The authors should consider expanding the discussion of strategies for improving tokenization methods in the related work section, such as [1] [2]. This would contribute to a more thorough understanding of the research landscape and enhance the paper's contributions.

[1] Havtorn J D, Royer A, Blankevoort T, et al. MSViT: Dynamic Mixed-Scale Tokenization for Vision Transformers[C]//Proceedings of the IEEE/CVF International Conference on Computer Vision. 2023: 838-848.
[2] Ronen T, Levy O, Golbert A. Vision Transformers with Mixed-Resolution Tokenization[C]//Proceedings of the IEEE/CVF Conference on Computer Vision and Pattern Recognition. 2023: 4612-4621.

**Questions:**

Have you considered conducting a comparative analysis between your superpixel tokenization method and mixed-scale tokenization approaches [1] [2]? Mixed-scale tokenization can also alleviate the limitations of regular grid partitioning methods to some extent, reducing information loss without introducing excessive computational overhead. What advantages do you offer in comparison to them?

---

> ### Author Response · Authors · 2023-11-17
>
> > This paper demonstrates a highly promising starting point by introducing the use of superpixel partitioning for irregular patch division, which seeks to address the inherent limitations of traditional grid-based partitioning methods in transformers.
>
> We sincerely thank the reviewer for their recognition of our contribution to the field.
>
> > However, the paper lacks an analysis of SPiT efficiency. [...] While Table 4 indicates that the superpixel algorithm used in this study performs comparably to state-of-the-art algorithms with substantially lower inference time, it is essential to explore whether the integration of superpixel tokens with subsequent transformer modules introduces a significant computational load. To address this, the authors should consider adding an additional set of comparative experiments to analyze the inference efficiency between ViT, RViT, and SPiT.
>
> We fully agree with the reviewer’s concerns that more dimensions for comparison will strengthen our paper. In response to the reviewers' concerns, in our revised submission, we included median images per second from training for all base capacity models to Table 1. This also relates to the reviewer’s next point about model performance, and we further address efficiency there.
>
> > The determination of hierarchical levels T is not clearly explained. To provide a more comprehensive understanding, the authors could include an additional set of ablation experiments to explore the implicit relationship between T settings and model performance.
>
> We acknowledge the importance of the reviewer’s point, and the **empirical experiments to determine the expected number of superpixel tokens over ImageNet1k have been added in Table F.1 and Figure F.1**, including uncertainty estimates. We include the results in Table F.1 for convenience.
>
> | Im.Size | $\mathbb E(\lvert \pi^{(1)} \rvert)$ | $\mathbb E(\lvert \pi^{(2)} \rvert)$ | $\mathbb E(\lvert \pi^{(3)} \rvert)$ | $\mathbb E(\lvert \pi^{(4)} \rvert)$ |
> |---------|------------------------------------|------------------------------------|------------------------------------|------------------------------------|
> | 224     | 11940.278(2.848)                   | 3155.512(0.808)                    | 794.650(0.209)                     | 197.411(0.052)                     |
> | 256     | 15496.020(3.786)                   | 4097.510(1.074)                    | 1031.727(0.277)                    | 256.051(0.071)                     |
> | 384     | 34084.297(9.188)                   | 9047.289(2.586)                    | 2287.822(0.669)                    | 567.69(0.172)                      |
>
> On average, $T=4$ produces nearly the exact same number of tokens in a ViT with patch size 16, while $T=3$ will obtain tokens equivalent on average to patch size 8. We expand the discussion in Section 2.2 and Appendix F to elucidate on the practical implications of this result, particularly in terms of inference. Briefly summarized, inference times are harder to theoretically estimate in data dependent tokenization schemes, since the number of tokens varies from image to image. However, the ablations in Table F.1 clearly show that **the number of tokens are; on average, essentially one-to-one to canonical ViT tokenization**. Then in light of the results in Table 4 (rev. Table 5), the effective increase in inference time will also be very close to the reported runtime of the SPiT tokenization step.
>
> Still, in light of the works by Havtorn et al. [1] and Ronen et al. [2] kindly suggested by the reviewer, there is an exciting research opportunity for extending our tokenization framework to be more effective, and we address this in our further work (Section 4). We emphatically thank the reviewer for their insightful contributions, and hope we were able to address them with due diligence.
>
> > The paper's summary of related work is relatively limited. The authors should consider expanding the discussion of strategies for improving tokenization methods in the related work section [...]
>
> We have significantly revamped the related work section thanks to the suggestions given by all the reviewers.  In the revised version, we touch upon the growing literature on tokenization in the transformers architectures and how **they have been focusing on changing the architectures to improve their efficiency**.  On the contrary, **we emphasize that we are working towards the complementary goal of decoupling the architectures** and bringing to the spotlight the vital role of the forgotten tokens. To contextualize these important works, we propose a taxonomy and illustrate the context in rev. Figure 4.

---

> ### Author Response · Authors · 2023-11-17
> **On mixed-scale tokenization**
>
> > Have you considered conducting a comparative analysis between your superpixel tokenization method and mixed-scale tokenization approaches [1] [2]? Mixed-scale tokenization can also alleviate the limitations of regular grid partitioning methods to some extent, reducing information loss without introducing excessive computational overhead.
> > What advantages do you offer in comparison to them?
>
> We thank the reviewer for the suggestions about highly relevant literature that we missed before. As previously touched upon, **the prospects of comparative analysis between superpixel tokenization and mixed-scale tokenization approaches is both welcome and appealing**, particularly since **both frameworks are posed to tackle similar limitations** in ViT architectures. On the one hand, mixed tokenization approaches encourage more effective partitioning for better throughput, efficiency, and better scaling for larger images, while superpixel tokenization seeks to solicit more granular partitions to improve interpretation, explainability, and improved performance for dense prediction tasks  (see recent addition in Section 3 and Appendix G). In our revised submission, we do indeed add a contrastive discussion on these approaches, given that they are largely commensurable with our own proposed framework, and **very much look in the same direction as our own work in expanding the tokenization process towards a modular cohesive whole**. We consider these as significant contributions to the overall research on modular tokenization, and graciously accept the reviewer’s suggestion. We meticulously expand our related work and discussion with these works in mind.

---

> > ### Comment · Reviewer_N399 · 2023-11-22
> >
> > Thank you for answer my questions carefully. I have no further issue and will keep my initial score.

---

> > > ### Author Response · Authors · 2023-11-22
> > > **Thank you for the response**
> > >
> > > Thank you for your time and your discerning response. We are very thankful for the acknowledgement, and all the insightful comments that has helped push our work to the next level.

---

### Official Review · Reviewer_ubVn · 2023-11-02

**Soundness:** 2 fair
**Presentation:** 3 good
**Contribution:** 2 fair
**Rating:** 5
**Confidence:** 4

**Summary:**

This paper presents a method that replaces the square patch tokenization by superpixel tokenization in vision transformers. The authors implemented a GPU-based superpixel method to serve as the tokens. The authors investigates the ViT using superpixel tokens for image classification tasks, and find that the proposed method is useful in some cases.

**Strengths:**

- The idea of replacing square tokens with superpixel tokens is straightforward and reasonable.
- Extensive experiments to analyze the proposed method.
- The proposed method has better explainability then vanilla square patch tokens.

**Weaknesses:**

- The experimental results are not very convincing. As stated in the paper "although we insist that the results are not significant enough to warrant any clear benefit for any framework in particular on classification tasks". It is not clear how useful is the proposed method in image classification and other tasks.
- Some sentences are not complete, e.g., "We hope that our work inspires more research into the" in page 9.

**Questions:**

The runtime analysis is missing. Would replacing the square patches with superpixels for tokens requires much extra runtime?
Do the superpixel tokens remain the same in each layer or are hierarchically grouped together?

---

> ### Author Response · Authors · 2023-11-17
>
> > The experimental results are not very convincing. As stated in the paper "although we insist that the results are not significant enough to warrant any clear benefit for any framework in particular on classification tasks". It is not clear how useful is the proposed method in image classification and other tasks.
>
> We appreciate the reviewer's feedback on our experimental results. Our intent is to showcase the adaptability of our framework across various datasets, highlighting that **no single tokenization strategy is universally superior**. For instance, our results demonstrate that coarse tokenization aligns better with low-resolution images like CIFAR, whereas finer tokenization shows advantages in high-resolution datasets like ImageNetReal. This variability underlines the context-dependent efficacy of different tokenization methods, rather than indicating an inherent weakness in any particular approach.
>
> We acknowledge that our paper may have given the impression of a lack of clear benefits for image classification tasks. However, **our aim was to illustrate that our tokenization framework can achieve comparable results to canonical ViT architectures** across a range of tasks. This is evidenced by our performance in standard classification tasks and further reinforced by our newly added experiments on unsupervised salient segmentation (Section 3, Appendix G), where **our framework performs competitively with state-of-the-art methods without additional training**. Our approach deliberately avoids significant deviations from the canonical ViT architecture to maintain a fair 'apples-to-apples' comparison between different tokenization strategies. This choice was made to highlight the potential of modular tokenization without obscuring the results with architecture-specific optimizations. Moreover, our work emphasizes the inherent interpretability and explainability of our approach, aspects we consider crucial for the practical application of vision models in real-world scenarios.
>
> Lastly, we must thank the reviewer for expressing their concerns regarding model performance, which contributed to us **expanding our experimental section with results on unsupervised salient segmentation in Appendix. G**, where we outperform SotA methods without any additional training of our framework. We hope our response speaks to your concerns regarding the benefits of our framework.
>
> > Some sentences are not complete, e.g., "We hope that our work inspires more research into the" in page 9.
>
> We sincerely thank the reviewer for the thorough evaluation.  We have remediated this issue, and further proofread the manuscript in the revised version.
>
> > The runtime analysis is missing.
>
> > Would replacing the square patches with superpixels for tokens requires much extra runtime?
>
> This is an astute observation, and we wholeheartedly agree that this is of central importance. In our original submission, we only briefly touch upon the efficiency of our tokenization scheme in Section 3.2. (Quality of superpixels) and Table 4 (rev. Table 5), where we show that our superpixel tokenizer achieves very strong inference runtimes compared to other state-of-the-art approaches (0.0047 sec/img.).  However, this does not address runtime costs during training. In response to the reviewer’s concern, **we have added comparative runtime analysis from the training stage in Table 1**, alongside effective parameter counts for all architectures.  We thank the reviewer for their contribution towards improving the quality of our work.
>
> > Do the superpixel tokens remain the same in each layer or are hierarchically grouped together?
>
> There seems to be a misunderstanding on how the superpixels are used in our framework, which we are hopefully able to clarify. The **superpixels are constructed hierarchically in the tokenization step** using a graph-based approach, as per Section 2.2. After the tokenization step, we extract features using our proposed approach (non-hierarchically), which are then fed to the general transformer architecture. The hierarchical nature of the superpixel algorithm provides ample opportunity for continued research with multi-scale feature hierarchies, and we discuss this as further work (Section 4), and this is a route we are currently exploring.

---

### Official Review · Reviewer_Apxd · 2023-11-05

**Soundness:** 2 fair
**Presentation:** 3 good
**Contribution:** 2 fair
**Rating:** 3
**Confidence:** 5

**Summary:**

This paper suggests using superpixels instead of patches as the input for vision transformers. It generates superpixels using a previous method with a different regularization term. Experiments show that superpixels have better interpretability than patches.

**Strengths:**

This paper has an interesting motivation to use superpixels instead of grids as the input for vision transformers. It also provides better interpretability by visualizing the attention maps. The paper gives the formulation of the proposed method and proves that the conventional ViT is a special case of it.

**Weaknesses:**

This paper has limited novelty, as previous works [1][2] have already explored the combination of superpixels and Transformers. The superpixels are created using only low-level and predefined features in the first layer, which may cause permanent errors. In contrast, [1][2] use the features learned by the network. This may explain why the performance on image classification is worse than the baseline.

[1] Vision Transformer with Super Token Sampling

[2] Superpixel Transformers for Efficient Semantic Segmentation

**Questions:**

Do you use a differentiable online tokenization process?

How does your method compare with conventional superpixel methods like SLIC, since you also use manually defined features?

---

> ### Author Response · Authors · 2023-11-17
>
> > This paper has an interesting motivation to use superpixels instead of grids as the input for vision transformers. It also provides better interpretability by visualizing the attention maps. The paper gives the formulation of the proposed method and proves that the conventional ViT is a special case of it.
>
> We thank the reviewer for praising our interesting motivation.
>
> > This paper has limited novelty, as previous works [1][2] have already explored the combination of superpixels and Transformers.
>
> Firstly, we emphasize that **our main contribution lies in the decoupling of tokenization from the core transformer architecture**, a distinctive approach from the previous works cited. *Unlike these works which necessitate significant architectural alterations* for different tokenization strategies, our method *does not alter the standard transformer architectures*. This compatibility and its implications are empirically demonstrated in orig. Table 3 of our paper (rev. Table 4). We note that **we already acknowledge and discuss the work by Huang et al. [1] in our original submission**, differentiating our approach. However, in light of the reviewers comments, we expand our discussion on this important work in our revised submission.
>
> Regarding Zhu et al. [2], it is pertinent to note that as far as we have found, the work was released as a preprint shortly after our submission deadline. **Given that our work thus predates Zhu et al. [2]**, which is considered novel by the reviewer, we believe **our work also merits recognition for its novelty on similar grounds**. We are committed to contributing novel insights to the field and hope this clarification addresses the reviewer's concerns regarding the novelty of our research.
>
> > The superpixels are created using only low-level and predefined features in the first layer, which may cause permanent errors.
>
> Unfortunately, the nature of these “permanent errors” is unclear to us, and we would greatly appreciate an elaboration. We hope to engage in dialogue to address the reviewer’s concerns and solicit constructive feedback for improving our work.
>
> > In contrast, [1][2] use the features learned by the network. This may explain why the performance on image classification is worse than the baseline.
>
> We must respectfully disagree with the reviewer. Evaluation of our results across tasks **indicates that superpixel tokenization is largely comparable to standard ViTs for classification** without optimized architectures or training regimes , and performance varies across evaluated datasets. We note an increase over ImageNetReal, and a general drop in CIFAR100, which favors square tokenization due to low resolution. Moreover, **we explicitly define our goal as demonstrating feasibility for modular tokenization strategies**, and **our results indicate that decoupling the tokens not only reproduces the original performance, but also allows for flexible ways of combining the tokens in future work** (see prev. Table 3, rev. Table 4).
>
> > Do you use a differentiable online tokenization process?
>
> We do not explore differentiable tokenization in this work. **Our focus is on the introduction of the idea of tokenization as a general modular component to the transformer architecture.** We agree with the reviewers overall sentiment that differentiable tokenization is an interesting prospect, but point out that such approaches must be balanced with potential costs in compute, as discussed in our original submission. Moreover, we highlight that the tokenization process in LLMs is generally not differentiable, but still highly effective. However, our own work towards this goal within our proposed framework is in progress, and we are excited to see continued work exploring tokenization in vision models.
>
> > How does your method compare with conventional superpixel methods like SLIC, since you also use manually defined features?
>
> We kindly direct the reviewer’s attention to prev. Table 4 (rev. Table 5.), **which explicitly compares our approach with SLIC, and shows that SLIC turns out to be less effective while providing lower quality superpixel partitions**. As for the discussion of “manually designed features”, we note that our proposed feature extraction is essentially designed to be commensurable and extendable to other ViT architectures to facilitate further research towards new modular tokenization strategies, aligning with what we set out to show.

---

### Author Response · Authors · 2023-11-17
**General Response**

We thank the reviewers for their constructive feedback, which has guided significant improvements in our manuscript. Our research primarily focuses on **decoupling tokenization from the transformer backbone**, aiming to harmonize with standard ViT architectures by way of modularity. This approach underlines the adaptability of tokenization strategies across various tasks, an aspect we believe is pivotal yet underexplored in current literature.

Our experiments establish **a fair comparison against well-known baselines without the inclusion of various optimizations to the backbone** (i.e., vanilla ViT architectures), highlighting the importance of understanding tokenizers for different tasks. **This controlled comparison is crucial for attributing observed performance disparities specifically to the tokenization techniques under scrutiny, and eliminates confounding factors from specialized architectures or training regimes.**

Addressing the concerns about the efficiency of our superpixel tokenizer, we have expanded our discussion and included additional empirical results in the revised manuscript. The new experiments in Section 3 and Appendix G, especially on unsupervised salient segmentation, underscore the robustness of our framework in state-of-the-art downstream tasks.

In response to reviewers valid concerns on our literature review, we have also **enriched our related work section**, offering a taxonomy of adaptive tokenization approaches and situating our work within this landscape. This includes an expanded discussion on mixed-scale tokenization and potential synergies with our approach, further contextualizing our contributions to the field.

As a result of the reviewers’ feedback, our revisions aim to clarify our methodology, strengthen our empirical evidence, and more clearly position our work within the broader context of vision transformer research. We believe these enhancements will solidify the understanding and appreciation of our paper's contributions.

**Overview of revisions:**
- In the interest of clarity, we reworked Sections 1-2 to emphasize the focus on decoupled adaptive tokenization with irregular patches, and discuss the benefits of such an approach to address the reviewers' concerns.
- In light of the reviewers concerns on baselines and results for segmentation tasks, we append experiments showing unsupervised salient segmentation results in the revised experiments (Sec. 3) and new Appendix G. We show that our framework outperforms existing SotA approaches for the $\max F_\beta$ metric across three datasets without additional training or post-processing.
- To address the reviewers concerns on efficiency, we append results for the number of images per second during training to Table 1, alongside parameter counts.
- Furthermore, we add estimates for the number of expected superpixels in Table F.1, demonstrating nearly exact average correspondence to patch sizes in standard ViT architectures over the full ImageNet1k training set. We explicitly derive the relation between patch sizes in standard ViTs and number of steps in SPiT, and show a near one-to-one correspondence, illustrated in Figure F.1. This shows that the level of granularity in SPiT tokenizers can be controlled by setting $T$ to achieve a desired level of granularity.
- We expand on the discussion in our related work section by
    - Suggesting a taxonomy of current adaptive tokenization approaches, and illustrate these (rev. Figure 4) to contextualize selected works.
    - We posit our work alongside recent mixed-scale tokenization approaches (Havtorn et al. Ronen et al.), whose modular approach coincides with our own while differing in overall goal. We also discuss strong potential for synergy between these approaches in the future work section.
    - Including the reference to ToME (Bolya et al.) whose approach improves throughput by merging tokens in transformer blocks, and expanding the discussion on SuperToken sampling (Huang et al.) to contrast these methods to modular tokenization.
- Lastly, we revise our future work to expand more on potential opportunities for our research, and expand on our current research focus continuing the work we present in this paper.

---

### Meta-Review · Area_Chair_Jz39 · 2023-12-07

**Metareview:**

This paper presents a tokenization for Transformer based on superpixels and decouples how the tokens are created and the feature extraction. The paper demonstrates competitive performance on classification, improved salient segmentation task. The main concern with the paper is improvement of the proposed method. Specifically, on classification task there are limited improvements, and the salient segmentation task is a somewhat niche task. The paper can be strengthened by demonstrating the effective of the approach on more popular benchmarks, e.g., instance segmentation. Also, the abstract is written is a somewhat misleading way "providing a space of semantically-rich models that can generalize across different vision tasks", however the they didn't actually show that the method works well across different visions tasks.

**Justification For Why Not Higher Score:**

The improvement in performance of the proposed approach is limiting in classification. The authors should demonstrate their claim in the abstract that the methods "generalize across different vision tasks" by conducting additional experiments on more competitive vision benchmarks, e.g., instance classification.

**Justification For Why Not Lower Score:**

N/A

---

### Decision · Program_Chairs · 2024-01-16

Reject